# NSGA-Net: A Multi-Objective Genetic Algorithm for Neural Architecture Search

## Abstract

This paper introduces NSGA-Net, an evolutionary approach for neural architecture search (NAS). NSGA-Net is designed with three goals in mind: (1) a NAS procedure for multiple, possibly conflicting, objectives, (2) efficient exploration and exploitation of the space of potential neural network architectures, and (3) output of a diverse set of network architectures spanning a trade-off frontier of the objectives in a single run. NSGA-Net is a population-based search algorithm that explores a space of potential neural network architectures in three steps, namely, a population *initialization* step that is based on prior-knowledge from hand-crafted architectures, an *exploration* step comprising crossover and mutation of architectures and finally an *exploitation* step that applies the entire history of evaluated neural architectures in the form of a Bayesian Network prior. Experimental results suggest that combining the objectives of minimizing both an error metric and computational complexity, as measured by FLOPS, allows NSGA-Net to find competitive neural architectures near the Pareto front of both objectives on two different tasks, object classification and object alignment. NSGA-Net obtains networks that achieve 3.72% (at 4.5 million FLOP) error on CIFAR-10 classification and 8.64% (at 26.6 million FLOP) error on the CMU-Car alignment task.

## 1 Introduction

Deep convolutional neural networks have been overwhelmingly successful at a variety of image analysis tasks. One of the key driving forces behind this success is the introduction of many CNN architectures such as, AlexNet (Krizhevsky et al., 2012), VGG (Simonyan & Zisserman, 2015), GoogLeNet (Szegedy et al., 2015), ResNet (He et al., 2016a), DenseNet (Huang et al., 2017) etc. in the context of object classification and Hourglass (Newell et al., 2016), and Convolutional Pose Machines (Wei et al., 2016a) in the context of object alignment. Concurrently, network designs such as MobileNet (Howard et al., 2017), XNOR-Net (Rastegari et al., 2016), BinaryNets (Courbariaux et al., 2016), LBCNN (Juefei-Xu et al., 2017) etc. have been developed with the goal of enabling real-world deployment of high performance models on resource constrained devices. These developments are the fruits of years of painstaking efforts and human ingenuity.

Neural architecture search (NAS) methods, on the other hand, seek to automate the process of designing network architectures. State-of-the-art reinforcement learning (RL) (Baker et al., 2017; Zhong et al., 2017; Zoph & Le, 2016; Zoph et al., 2017; Hsu et al., 2018a; Cai et al., 2018; Pham et al., 2018) and evolutionary algorithm (EA) (Miikkulainen et al., 2017; Real et al., 2017; 2018; Xie & Yuille, 2017; Kim et al., 2017; Liu et al., 2018; Elsken et al., 2018; Liu et al., 2017; Dong et al., 2018) approaches for NAS focus on the single objective of minimizing an error metric on a task and cannot be easily adapted to handle minimizing multiple, possibly conflicting, objectives. Methods like (Real et al., 2017) and (Zoph et al., 2017) are inefficient in their use of their search space and require 3,150 and 2,000 GPU days, respectively. Furthermore, most state-of-the-art approaches search over a single computation block, similar to an Inception block (Szegedy et al., 2015), and repeat it as many times as necessary to form a complete network.

In this paper, we present NSGA-Net, a genetic algorithm for NAS to address the aforementioned limitations of current approaches. The salient features of NSGA-Net are, (1) **Multiobjective Optimization**: Real-world deployment of NAS models is seldom guided by a single objective and often has to balance between multiple, possibly competing, objectives. For instance, we seek to maximize

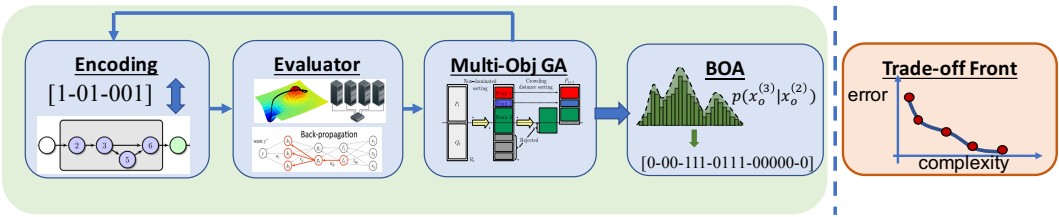

Figure 1: Overview of the stages of NSGA-Net. Networks are represented as bit strings, trained through gradient descent, ranking and selection by NSGA-II, search history exploitation through BOA. Output is a set of networks that span a range of complexity and error objectives.

performance on compute devices that are often constrained by hardware resources in terms of power consumption, available memory, available FLOPS, and latency constraints, to name a few. NSGA-Net is explicitly designed to optimize such competing objectives. (2) **Complete Architecture Search Space**: The search space for most existing methods is restricted to a block that is repeated as many times as desired. In contrast, NSGA-Net searches over the entire structure of the network. This scheme overcomes the limitations inherent to repeating the same computation block throughout an entire network, namely, that a single block may not be optimal for every application and it is desirable to allow NAS to discover architectures with different blocks in different parts of the network. (3) **Non-Dominated Sorting**: The core component of NSGA-Net is the Non-Dominated Sorting Genetic Algorithm II (NSGA-II) (Deb et al., 2000), a multi-objective optimization algorithm that has been successfully employed for solving a variety of multiobjective problems (Tapia & Coello, 2007; Pedersen & Yang, 2006). Here, we leverage it's ability to maintain a diverse trade-off frontier between multiple, possibly conflicting, objectives, thereby resulting in a more effective and efficient exploration of the search space, (4) **Crossover**: To fully utilize the diverse frontier of solutions provided by non-dominated sorting, we employ crossover (in addition to mutation) to combine networks with desirable qualities across multiple objectives, and finally (5) **BOA**: We construct and employ a Bayesian Network inspired by the Bayesian Optimization Algorithm (BOA) (Pelikan et al., 1999) to fully utilize the promising solutions present in our search history archive and the inherent correlations between the layers of the network architecture.

We demonstrate the efficacy of NSGA-Net on two tasks: image classification (CIFAR-10 (Krizhevsky & Hinton, 2009)) and object alignment or key-point prediction (CMU-Car (Boddeti et al., 2013)). For both tasks we minimize two objectives: an error metric and computational complexity. Here, computational complexity is defined by the number of floating-point operations or multiply-adds (FLOPS). Experimentally, we observe NSGA-Net can find a set of network architectures containing solutions that are significantly better than hand-crafted methods in both objectives while being competitive in single objectives to state-of-the-art NAS approaches. Furthermore, by fully utilizing a population of networks through crossover and utilization of the search history, NSGA-Net explores the search space efficiently and requires less computation time for searching than competing methods.

## 2 RELATED WORK

Recent research efforts in NAS have produced a plethora of methods to automate the design of networks. Broadly speaking, these methods can be divided into EA approaches and RL approaches—with a few methods not falling into either category. Due to space constraints, we provide a comprehensive overview of relevant methods and a table summarizing their contributions in Appendix D. Here, we focus on multiobjective methods, and relate NSGA-Net to recent contributions.

Kim et al. (2017) presented NEMO, one of the earliest multiobjective approaches involving neural networks. NEMO used NSGA-II (Deb et al., 2000) to minimize error and inference time of a network and searched over the space of the number of output channels from each layer within a restricted space of 7 different architectures. In contrast, NSGA-Net seeks to use NSGA-II to minimize error and computational complexity (as measured by FLOPS), while searching over the vast space of possible network architectures but with fixed hyperparameters. Dong et al. (2018) proposed PPP-Net as a multiobjective extension to the progressive NAS method in (Liu et al., 2017) that uses a predictive model to choose promising networks to train in an effort to reduce computational strain. PPP-Net

differs from NSGA-Net by not using crossover and the use of this predictive model. Elsken et al. (2018) present the LEMONADE method which is formulated to develop networks with low error and number of parameters through custom designed approximate network morphisms (Wei et al., 2016b). This allows newly generated networks to share parameters with their forerunners, obviating network training from scratch. NSGA-Net differs from LEMONADE in terms of the encoding scheme, the network morphisms as well in the selection scheme. NSGA-Net relies on, (1) genetic operations like mutation and crossover, encouraging population diversity, instead of custom network morphism operations, and (2) NSGA-II, rather than novelty-based sampling as the selection scheme in LEMONADE, with the former affording a more efficient exploration of the search space. Finally, we highlight MONAS, presented by Hsu et al. (2018a). This approach is one of the first—if not the only—RL NAS method to use a multiobjective scheme. MONAS searches the same space that is presented in (Zoph et al., 2017) while using a reward signal to optimize a recurrent neural network to generate convolutional networks. The reward signal however is a linear combination of accuracy and energy consumption of a network. However, it is well known in multiobjective optimization literature that a simple linear combination of objectives suffers from a number of drawbacks, including sub-optimality (Koski, 1985) of the search (see Section 3 for a more detailed discussion).

## 3 PROPOSED APPROACH

Compute devices are often constrained by hardware resources in terms of power consumption, available memory, available FLOPS, and latency constraints. Hence, real-world design of DNNs are required to balance multiple, possibly competing, objectives (e.g., predictive performance and computational complexity). Often, when multiple design criteria are considered simultaneously, there may not exist a single solution that performs optimally in all desired criteria, especially with competing objectives. Under such circumstances, a set of solutions that provides the entire trade-off information between the objectives is more desirable. This enables a practitioner to analyze the importance of each criterion, depending on the application, and to choose an appropriate solution on the trade-off frontier. We propose NSGA-Net, a genetic algorithm based architecture search method to automatically generate a set of DNN architectures that approximate the Pareto-front between performance and complexity on classification and regression tasks. The rest of this section describes the design principles, encoding scheme, and main components of NSGA-Net in detail.

### 3.1 DESIGN PRINCIPLES

**Population Based Methods:** The two main approaches to obtaining an efficient set of trade-off solutions are: i) classical point-based methods like a weighted sum of the objective functions (e.g., $\alpha f_1(\boldsymbol{x}) + (1 - \alpha)f_2(\boldsymbol{x})$); and ii) population-based methods like genetic algorithms. Despite the well characterized convergence properties of point-based methods, they often present the following challenges as illustrated in Fig.2: 1) They require prior knowledge of the range of values each objective function can take for appropriate normalization before weighting, else, the solutions are often biased towards the objective with large values; 2) weighted combinations of objectives can only find solutions on convex regions of the Pareto-front and are incapable of discovering solutions on concave parts of the Pareto-front (Koski, 1985); finally 3) obtaining each solution on the Pareto-front requires the repetition of the entire search procedure for each combination weight $\alpha$.

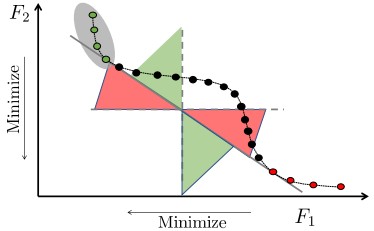

Figure 2: (1) Black solutions on the concave segment of the frontier cannot be achieved through weighted sum; (2) If scale of $f_2(\boldsymbol{x})$ is much larger than $f_1(\boldsymbol{x})$, weighted sum solutions are biased towards the shaded grey region; (3) green and red solutions are possible from linear combination, but obtaining each solution requires an independent run.

On the other hand, population-based methods, like genetic algorithms, offer a flexible, viable alternative to find multiple efficient trade-off solutions in one execution. A population-based method is capable of introducing an *implicit* parallel search by processing multiple solution candidates jointly at each iteration. However, optimizing with $N$ population members processes $O(N^3)$ sub-regions of the search space in parallel (Goldberg, 1989; Holland, 1975). The efficiency afforded by such parallelism cannot be matched by point-based methods like weighted combinations. Recent studies

Figure 3: **(a)** An example of how non-dominated sorting ranks solutions into different frontiers based on objectives, $f_1(\cdot)$ and $f_2(\cdot)$. **(b)** The environmental selection procedure, in which created offspring, $Q_t$, networks get merged with parent population, $P_t$, to create $R_t$ from which the "better" half survives into the next parent population, $P_{t+1}$. Here, better is defined as a hierarchical preference of rank and crowding distance. **(c)** A pictorial description of crowding distance. All points in this figure are non-dominated. To select a subset of them, we must rely on crowding distance. Points in the blue region are preferred to the orange region since they are further from their nearest neighbors.

show that population-based methods can successfully solve problems with millions (Chicano et al., 2017) or billions (Deb & Myburgh, 2017) of variables while classical point-based methods, like branch-and-bound, fail even with hundreds of variables.

**Multi-Criterion Based Selection:** The general form of a multiobjective optimization problem is

$$\min_{\boldsymbol{x}} \left\{ f_1(\boldsymbol{x}), \ldots, f_M(\boldsymbol{x}) \right\} \tag{1}$$

where $f_i(\cdot)$ are the criterion that we wish to optimize and $\boldsymbol{x}$ is the representation of a neural network architecture (described in Section 3.2). For the aforementioned problem, given solutions $\boldsymbol{x}_1$ and $\boldsymbol{x}_2$, $\boldsymbol{x}_1$ is said to dominate $\boldsymbol{x}_2$ (i.e., $\boldsymbol{x}_1 \preceq \boldsymbol{x}_2$) if both of the following conditions are satisfied,

1. $\boldsymbol{x}_1$ is no worse than $\boldsymbol{x}_2$ for all objectives ($f_i(\boldsymbol{x}_1) \leq f_i(\boldsymbol{x}_2) \ \forall i \in \{0, \ldots, M\}$)
2. $\boldsymbol{x}_1$ is strictly better than $\boldsymbol{x}_2$ in at least one objective ($\exists \ i \in \{0, \ldots, M\} \mid f_i(\boldsymbol{x}_1) < f_i(\boldsymbol{x}_2)$)

Therefore, a solution $\boldsymbol{x}_i$ is non-dominated if these conditions hold for all $\boldsymbol{x}_j$ and $j \neq i$.

The core of NSGA-Net is a selection criterion that leverages non-dominated solutions. Specifically, given a population of network architectures $\{\boldsymbol{x}_1, \ldots, \boldsymbol{x}_n\}$ and their fitness functions $\{f_1\boldsymbol{x}_i, \ldots, f_M(\boldsymbol{x}_i)\}$, the ranking and selection procedure consists of two stages: (1) non-dominated solutions are selected over dominated solutions; (2) explicitly ranking of solutions that are diverse w.r.t. to the trade-off between the objectives higher than solutions that are "crowded" on the trade-off front (see Fig. 3c) i.e., how close a given solution is to its neighbors in the objective space. We adopt the non-domination ranking and crowded-ness measurements proposed by Deb et al. (2000).

The non-domination ranking indicates the front number that a solution belongs to; these fronts are composed by the set of non-dominated solutions at the current search iteration. An example of the non-domination fronts and non-domination ranking is shown in Fig. 3a and Fig. 3b, respectively. It's worth noting that both non-domination ranking and crowded-ness are relative measurements which need to be re-assessed when new solutions are created.

**Elitist-preserving**: This term refers to the fact that the best solution (in terms of objective values and crowded-ness) in the parent population will always be carried on to the next population. This allows the previous best solution to always have a chance to share its genetic information with the next generation, without the risk of losing the information to the newly generated child population. As a consequence of this scheme, the best solution encountered during the entirety of the search will always be present in the final population.

### 3.2 ENCODING

Genetic algorithms, like any other biologically inspired search methods, do not directly operate on *phenotypes*. From the biological perspective, we may view the DNN architecture as a phenotype, and the representation it is mapped from as its *genotype*. As in the natural world, genetic operations like

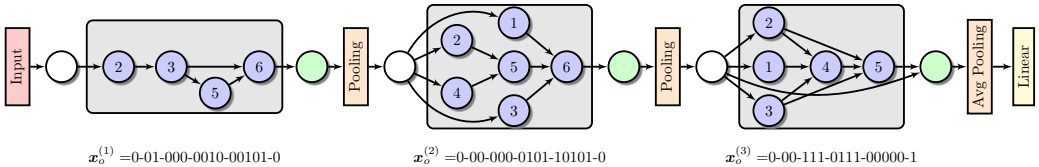

$x_o^{(1)}$ =0-01-000-0010-00101-0     $x_o^{(2)}$ =0-00-000-0101-10101-0     $x_o^{(3)}$ =0-00-111-0111-00000-1

Figure 4: **Encoding:** Illustration of a classification network encoded by $\mathbf{x} = (\mathbf{x_p}, \mathbf{x_o})$, where $x_o$ is the operations at a phase (gray boxes, each with a possible maximum of 6 nodes) and $x_p$ is the processing resolution path (orange boxes that connect the phases). In this example the path, $\mathbf{x_p}$, is fixed based on prior knowledge of successful approaches. The phases are described by the bit string $\mathbf{x_o}$ which is formatted for readability above. The bits are grouped by dashes to describe what node they control. See Appendix E for detailed description of the encoding schemes.

crossover and mutation are only carried out in the genotype space; such is the case in NSGA-Net as well. We refer to the interface between the genotype and the phenotype as *encoding* in this paper.

Most existing CNN architectures can be viewed as a composition of computational blocks (e.g. ResNet blocks (He et al., 2016a), DenseNet block (Huang et al., 2017), and Inception block (Szegedy et al., 2015), etc.) and resolution of information flow path. For example, down-sampling is often used after a computational block to reduce the information resolution going into the next computational block in networks designed for classification tasks. In NSGA-Net, each computational block, referred to as a *phase*, is encoded using the method presented by Xie & Yuille (2017), with the small change of adding a bit to represent a skip connection. And we name it as the *operation* encoding $\mathbf{x_o}$. To handle the resolutions of the information flowing paths, we present a novel encoding, named the *path* encoding $\mathbf{x_p}$, that takes inspiration from the Hourglass (Newell et al., 2016) architecture. Hence, each DNN architecture in NSGA-Net is presented as a tuple $\mathbf{x} = (\mathbf{x_p}, \mathbf{x_o})$. Though NSGA-Net operates in the genotype space, the performance of each solution is assessed based on its phenotype.

### 3.2.1 OPERATION ENCODING $\mathbf{x_o}$

Unlike most of the hand-crafted and NAS generated architectures, we do not repeat the same phase (computational block) to construct a network. Instead, the operations of a network are encoded by $\mathbf{x_o} = \left( \mathbf{x_o^{(1)}}, \mathbf{x_o^{(2)}}, \dots, \mathbf{x_o^{(n_p)}} \right)$ where $n_p$ is the number of phases. Each $\mathbf{x_o^{(i)}}$ encodes a directed acyclic graph consisting of $n_o$ number of nodes that describes the operation within a phase using a binary string. Here, a *node* is a basic computational unit, which can be a single operation like convolution, pooling, batch-normalization or a sequence of operations. This encoding scheme offers a compact representation of the network architectures in genotype space, yet is flexible enough that many of the computational blocks in hand-crafted networks can be encoded, e.g VGG (Simonyan & Zisserman, 2015), ResNet (He et al., 2016a) and DenseNet (Huang et al., 2017). Figure 4 shows an example of the operation encoding and more details are provided in Appendix E.1.

### 3.2.2 PATH ENCODING $\mathbf{x_p}$

In NSGA-Net, the path encoding, $\mathbf{x_p} = \left( \mathbf{x_p^{(1)}}, \mathbf{x_p^{(2)}}, \dots, \mathbf{x_p^{(n_p)}} \right)$, is a $n_p$-dimensional integer vector whose entries are in the range $[-r, r]$, $r$ is the original input information resolution (e.g. for CIFAR-10, $r = 32$) and $n_p$ is the number of phases (computational blocks) in the network. Each $\mathbf{x_p}^{(i)}$ indicates the stride value for the pooling operation after phase $i$, where positive and negative values denote up-sampling and down-sampling respectively, and zero encodes no change in resolution.

### 3.2.3 SEARCH SPACE

The search domain, $\Omega_{\mathbf{x_o}}$, defined by our operation encoding $\mathbf{x_o}$ consists of $n_p \times 2^{n_o(n_o-1)/2+1}$ strings. And the search domain, $\Omega_{\mathbf{x_p}}$, defined by our path encoding $\mathbf{x_p}$ consists of $(2r)^{n_p}$ combinations. Hence, the total search domain in the genotype space is:

$$\Omega_{\mathbf{x}} = \Omega_{\mathbf{x_p}} * \Omega_{\mathbf{x_o}} = (2r)^{n_p} \times n_p \times 2^{n_o(n_o-1)/2+1}$$

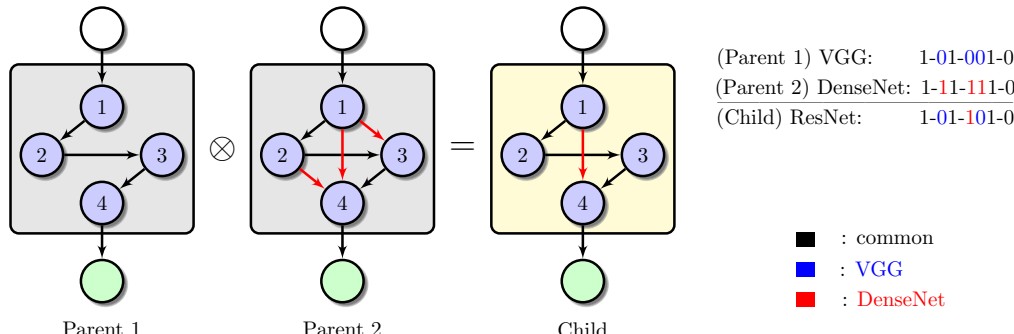

Figure 5: **Crossover Example:** A crossover (denoted by ⊗) of a VGG-like structure with a DenseNet-like structure may result in a ResNet-like network. In the figure, red and blue denotes connections that are unique to VGG and DenseNet respectively, and black shows the connections that are common to both parents. All black bits are retained in the final child encoding, and only the bits that are not common between the parents can potentially be selected at random from one of the parent.

where $n_p$ is the number of phases (computational blocks), $n_o$ is the number of nodes (computational units) in each phase, and $r$ is the original input information resolution. However, for computationally tractability, we constrain the search space in two ways: **(1)** each node (computational unit) in a phase (computational block) carriers the same sequence of operations, i.e. a $3 \times 3$ convolution followed by batch-normalization and ReLU; **(2)** we restrict the search space to only the operation encoding i.e, $\mathbf{x_o}$ and fix the path encoding to $2 \times 2$ max-pooling with stride 2, i.e., $\mathbf{x_p^{(i)}} = -2$ and use a global averaging pool before classification layer.

It is worth noting that, as a result of nodes in each phase having identical operations, the encoding between genotype and phenotype is a many-to-one mapping. Given the prohibitive computational expense required to train each network architecture before its performance can be assessed, it is essential to avoid evaluating genomes that decode to the same architecture. We develop an algorithm to quickly and approximately identify these duplicate genomes (see Appendix E.3 for details).

### 3.3 SEARCH PROCEDURE

NSGA-Net is an iterative process in which initial solutions are made gradually better as a group, called *population*. In every iteration, the same number of offspring (new network architectures) are generated from parents selected from the population. Each population member (including both parents and offspring) compete for both survival and reproduction (becoming a parent) in the next iteration. The initial population may be generated randomly or guided by prior-knowledge (e.g. seeding the hand-crafted network architectures into the initial population). The overall NSGA-Net search proceeds in two sequential stages, an *exploration* and *exploitation*.

#### 3.3.1 EXPLORATION

The goal of this stage is to discover diverse ways of connecting nodes to form a phase (computational block). Genetic operations, crossover[1] and mutation, offer an effective mean to realize this goal.

**Crossover:** The *implicit* parallelism of population-based search approaches can be unlocked when the population members can effectively share (through crossover) building-blocks (Holland, 1975). In the context of NAS, a phase or the sub-structure of a phase can be viewed as a building-block. We design a homogeneous crossover operator, which takes two selected population members as parents, to create offspring (new network architectures) by inheriting and recombining the building-blocks from parents. The main idea of this crossover operator is to 1) preserve the common building-blocks shared between both parents by inheriting the common bits from both parents' binary bit-strings; 2) maintain, relatively, the same complexity between the parents and their offspring by restricting the

---

[1]Population-based search without crossover, using mutation only, ceases to be a population-based method and is equivalent to running a point-based search individually with different initializations.

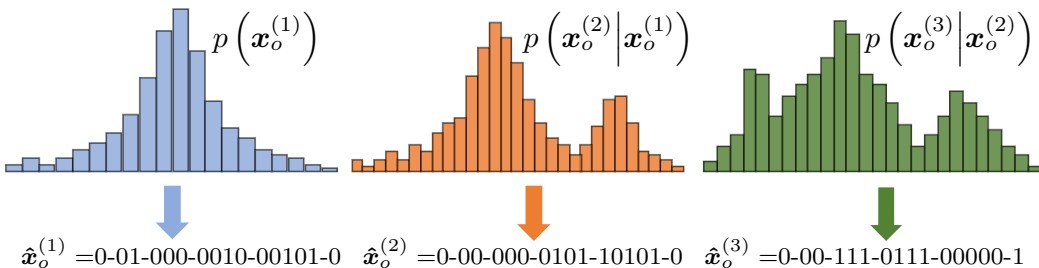

$\hat{\boldsymbol{x}}_o^{(1)}$ =0-01-000-0010-00101-0   $\hat{\boldsymbol{x}}_o^{(2)}$ =0-00-000-0101-10101-0   $\hat{\boldsymbol{x}}_o^{(3)}$ =0-00-111-0111-00000-1

Figure 6: **Exploitation:** Sampling from the Bayesian Network (BN) constructed by NSGA-Net. The histograms represent estimates of the conditional distributions between the network structure between the phases explored during the exploration step and updated during the exploitation step (i.e., using the population archive). During exploitation, networks are constructed by sampling phases from the BN. Fig. 4 shows the architectures that the sampled bit strings, $\{\hat{\boldsymbol{x}}_o^{(1)}, \hat{\boldsymbol{x}}_o^{(2)}, \hat{\boldsymbol{x}}_o^{(3)}\}$ decode to.

number of "1" bits in the offspring's bit-string to lie between the number of "1" bits in both parents. An example of the crossover operator is provided in Figure 5.

**Mutation:** To enhance the diversity (having different network architectures) of the population and the ability to escape from local optima, we use a bit-flipping mutation operator, which is commonly used in binary-coded genetic algorithms. Due to the nature of our encoding, a one bit flip in the genotype space could potentially create a completely different architecture in the phenotype space. Hence, we restrict the number of bits that can be flipped to be at most one for each mutation operation.

### 3.3.2 Exploitation

The exploitation stage follows exploration in NSGA-Net. The goal of this stage is to exploit the archive of solutions explored in the previous stage. The exploitation step in NSGA-Net is heavily inspired by the Bayesian Optimization Algorithm (BOA) (Pelikan et al., 1999) which is explicitly designed for problems with inherent correlations between the optimization variables. In the context of our NAS encoding, this translates to correlations in the blocks and paths across the different phases. Exploitation uses past information across all networks evaluated to guide the final part of the search. More specifically, say we have a network with three phases, namely $\mathbf{x_o^{(1)}}$, $\mathbf{x_o^{(2)}}$, and $\mathbf{x_o^{(3)}}$. We would like to know the relationship of the three phases. For this purpose, we construct a Bayesian Network (BN) relating these variables, modeling the probability of networks beginning with a particular phase $\mathbf{x_o^{(1)}}$, the probability that $\mathbf{x_o^{(2)}}$ follows $\mathbf{x_o^{(1)}}$, and the probability that $\mathbf{x_o^{(3)}}$ follows $\mathbf{x_o^{(2)}}$. In other words we estimate the distributions $p\left(\mathbf{x_o^{(1)}}\right)$, $p\left(\mathbf{x_o^{(2)}}|\mathbf{x_o^{(1)}}\right)$, and $p\left(\mathbf{x_o^{(3)}}|\mathbf{x_o^{(2)}}\right)$ by using the population history, and update these estimates during the exploitation process. New offspring solutions are created by sampling from this BN. Figure 6 shows a pictorial depiction of this process.

## 4 Experiments

In this section, we demonstrate the efficacy of NSGA-Net to automate the NAS process for two tasks: image classification and object alignment (regression). Due to space constraints, we only present classification results here and present all the regression results in Appendix G.

### 4.1 Performance Metrics

We consider two objectives to guide NSGA-Net based NAS, namely, classification error and computational complexity. A number of metrics can serve as proxies for computational complexity: number of active nodes, number of active connections between the nodes, number of parameters, inference time and number of multiply-add operations (FLOPs) needed to execute the forward pass of a given network. Our initial experiments considered each of these different metrics. We concluded from extensive experimentation that inference time cannot be estimated reliably due differences and inconsistencies in computing environment, GPU manufacturer and temperature etc. Similarly, the

number of parameters, active connections or active nodes only relate to one aspect of computational complexity. In contrast, we found an estimate of FLOPs to be a more accurate and reliable proxy for network complexity. See Appendix F for more details. Therefore, classification error and FLOPs serve as the twin objectives for selecting networks.

For the purpose of quantitatively comparing different multi-objective search methods or different configuration setups of NSGA-Net, we use the hypervolume (HV) performance metric, which calculates the dominated area (hypervolume in the general case) from the a set of solutions (network architectures) to a reference point which is usually an estimate of the nadir point—a vector concatenating worst objective values of the Pareto-frontier. It has been proved that the maximum HV can only be achieved when all solutions are on the Pareto-frontier (Fleischer, 2003). Hence, the higher the HV measures, the better solutions that are being found in terms of both objectives. See Figure 7 for a hypothetical example.

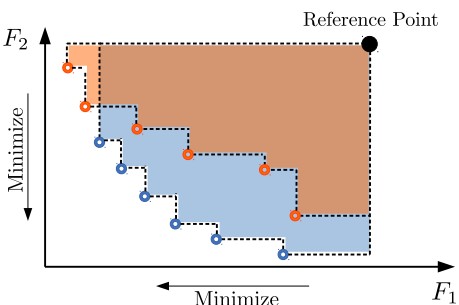

## 4.2 Implementation Details

**Dataset:** We consider the CIFAR-10 (Krizhevsky et al.) dataset for our classification task. We split the original training set (80%-20%) to create our training and validation sets. The original CIFAR-10 testing set is only utilized at the conclusion of the search to obtain the test accuracy for the models on the final trade-off front. **NSGA-Net hyper-parameters:** We set the number of phases $n_p$ to three and the number of nodes in each phase $n_o$ to six. We also fix the path encoding $\mathbf{x_p} = [-2, -2, -8]$, which decodes to having a max-pooling with stride 2 after the first and the second phase, and a global average pooling layer after the last phase. The initial population is generated by uniform random sampling. The population size is 40 and the number of iterations is 20 for the exploration stage. During exploitation, we

Figure 7: An illustration of the computation of HV in a hypothetical bi-objective scenario with two experiments. The points on the trade-off frontier and a chosen reference point form a bounding box from which HV is computed. Here, the blue experiment would be considered a more desirable result since it covers more of the objective space, even though all its solutions may not dominate all the points shown in orange.

reduce the number of iterations by half. Hence, a total of 1,200 network architectures are searched by NSGA-Net. **Network training:** For training each generated network architecture, we use standard stochastic gradient descent (SGD) and a cosine annealing learning rate schedule (Loshchilov & Hutter, 2016). Our initial learning rate is 0.025 and we train for 25 epochs, which takes about 9 minutes on a NVIDIA 1080Ti GPU implementation in PyTorch (Paszke et al., 2017).

## 4.3 Results

Figure 8b shows the bi-objective frontiers obtained by NSGA-Net through the various stages of the search, clearly showcasing a gradual improvement of the whole population. Figure 8c shows two metrics: normalized HV and offspring survival rate, through the different generations of the population. The monotonic increase in the former suggests that a better set of trade-off network architectures have been found over the generations. The monotonic decrease in the latter metric suggests that, not surprisingly, it is increasingly difficult to create better off-springs (than their parents). We can use a threshold on the offspring survival rate as a potential criterion to terminate the current stage of the search process and switch between the exploration and exploitation.

To compare the trade-off front of network architecture obtained from NSGA-Net to other hand-crafted and search-generated architectures, we pick the network architectures with the lowest classification error from the final frontier (the dot in the lower right corner on the green curve in Figure 8b) and extrapolate the network by increasing the number of filters of each node in the phases, and train with the entire official CIFAR-10 training set. The chosen network architecture results in 3.85% classification error on the CIFAR-10 testing set with 3.3 Millions of parameters and 1290

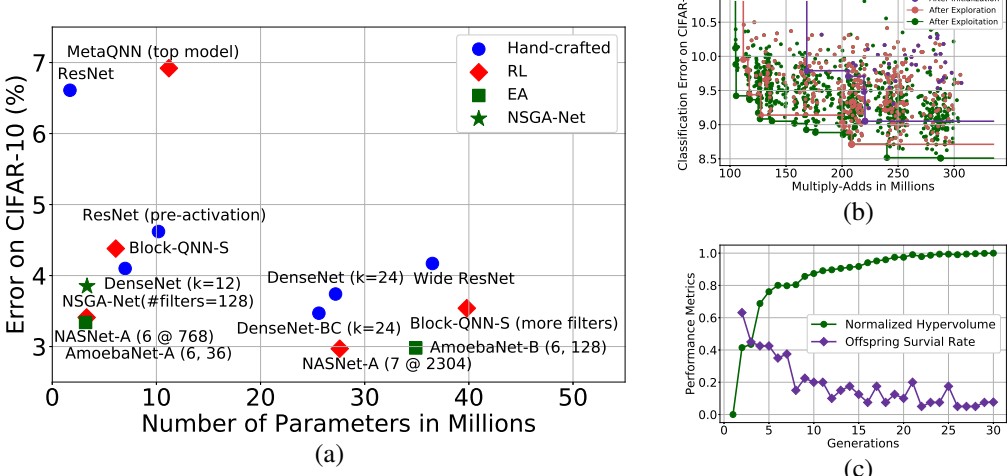

Figure 8: (a) Accuracy Comparison (b) Progression of trade-off frontiers at each stage of NSGA-Net. (c) Generational normalized hypervolume and survival rate of the offspring network architectures.

Table 1: Multiobjective methods for CIFAR-10 (best accuracy for each method)

| Method | Error | Other Objective | Compute |
|--------|-------|-----------------|---------|
| PPP-Net (Dong et al., 2018) | 4.36 | FLOPs or Params or Inference Time | Nvidia Titan X |
| MONAS (Hsu et al., 2018a) | 4.34 | Power | Nvidia 1080Ti |
| NSGA-Net (Ours) | 3.85 | FLOPS | Nvidia 1080Ti 8 GPU Days |

MFLOPs (green star in Figure 8a). Table 4 provides a summary that compares NSGA-Net with other multi-objective NAS methods. We include the comprehensive comparison table in Appendix G.

## 4.4 ABLATION STUDIES

Here, we first present results comparing NSGA-Net with uniform random sampling (RSearch) from our encoding as a sanity check. It's clear from Figure 9a that much better set of network architectures are obtained using NSGA-Net. Then we present additional results to showcase the benefits of the two main components of our approach: crossover and Bayesian network based offspring creation. **Crossover Operator:** Current state-of-the-art NAS search results (Liu et al., 2018; Real et al., 2018) using evolutionary algorithms use mutation alone with enormous computation resources. We quantify the importance of crossover operation in an EA by conducting the following small-scale experiments on different datasets, including MNIST, SVHN, and CIFAR-10. From figs. 9b and 9c, we observe that crossover helps achieve a better trade-off frontier and performance w.r.t. to both criteria across these datasets. **Bayesian Network (BN) based Offspring Creation:** Here we quantify the benefits of the exploitation stage i.e., off-spring creation by sampling from BN. We uniformly sampled 120 network architectures each from our encoding and from the BN constructed on the population archive generated by NSGA-Net at the end of exploration. The architectures sampled from the BN dominate (see Fig.9d) all network architectures created through uniform sampling.

## 4.5 DISCUSSION

We analyze the intermediate solutions of our search and the trade-off frontiers and make some observations. Upon visualizing networks, like the one in Figure 4, we observe that as network

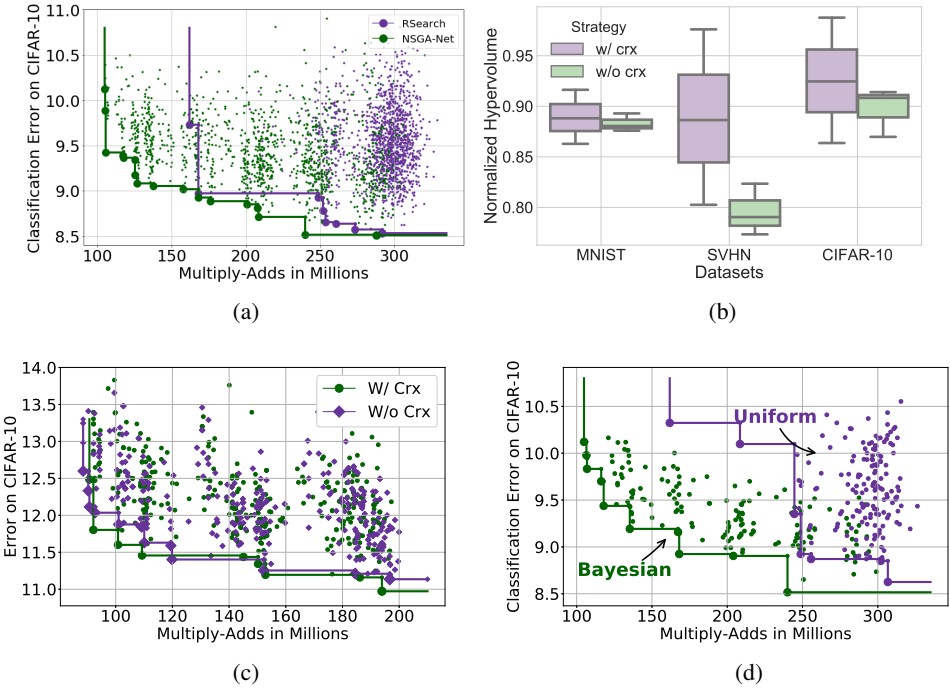

Figure 9: (a) Trade-off frontier comparison between random search and NSGA-Net. (b) Normalized hypervolume statistics over five runs. Higher normalized hypervolume implies better performance on both criteria. (c) Trade-off frontier comparison with and without crossover. (d) Comparison between sampling from uniformly from the encoding space and the Bayesian Network constructed from NSGA-Net exploration population archive.

complexity decreases along the front, the search process gravitates towards reducing the complexity by minimizing the amount of processing at higher image resolutions i.e., remove nodes from the phases that are closest to the input to the network. As such, NSGA-Net outputs a set of network architectures that are optimized for wide range of complexity constraints. On the other hand, approaches that search over a single repeated computational block can only control the complexity of the network by manually tuning the number of repeated blocks used. Therefore, NSGA-Net provides a more fine-grained control over the two objectives as opposed to the control afforded by arbitrary repetition of blocks. Moreover, some objectives, for instance susceptibility to adversarial attacks, may not be easily controllable by simple repetition of blocks. Figure 20 in the Appendix shows a subset of networks discovered on the trade-off frontier for CIFAR-10.

## 5 CONCLUSION

This paper presented NSGA-Net, a multi-objective evolutionary approach for neural architecture search. NSGA-Net affords a number of practical benefits: (1) the design of neural network architectures that can effectively optimize and trade-off multiple, possibly, competing objectives, (2) advantages afforded by population based methods being more effective than optimizing weighted linear combination of objectives, (3) more efficient exploration and exploitation of the search space through a novel crossover scheme and leveraging the entire search history through BOA, and finally (4) output a set of solutions spanning a trade-off front in a single run. Experimentally, by optimizing both prediction performance and computational complexity NSGA-Net finds networks that are significantly better than hand-crafted networks on both objectives and is compares favorably to other state-of-the-art single objective NAS methods for classification on CIFAR-10 and object alignment (regression) on CMUCars.

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

# Appendix

## A  UTILITY OF NSGA-NET: ROBUSTNESS AGAINST ADVERSARIAL ATTACK

Hand designing a neural network architecture for a given task is an intensive and time consuming process. When the task at hand, target datasets or the goal of the network architecture changes, the whole complex process needs to be repeated again. In reality, one simply modifies existing network architectures developed on other tasks and adopts them for the task at hand. This procedure while successful, has limited scalability.

In this section, we provide another proof-of-concept example of the utility of NSGA-Net. We consider the task of finding neural architectures that not only demonstrate high accuracy for image classification but simultaneously be robust against adversarial attacks. Our choice of this problem is motivated by the following factors: (i) To demonstrate the utility of NSGA-Net on a task that is not easily controlled by hand. Single objective based NAS methods are able to control a second objective (computational complexity) by manually controlling the number of repeated blocks or number of filters etc. Robustness to adversarial attacks on the other hand cannot be achieved in the same way. Furthermore, we conjecture that these two objectives are competing and hence are a good test of the utility of NSGA-Net, (ii) Explore the effect of network architecture on their robustness to adversarial attacks.

### A.0.1  EXPERIMENT SETUP

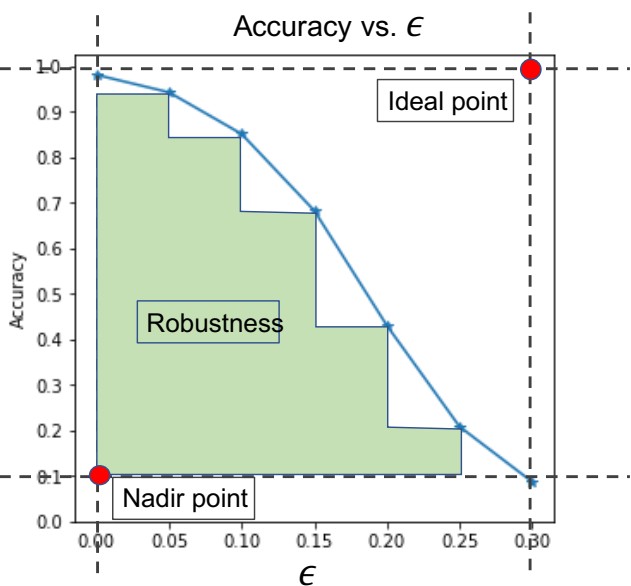

Figure 10: **Robustness Objective:** We define a robustness objective under the FGSM Goodfellow et al. (2014) attack as follows: 1) obtain classification accuracy on adversarial images generated by FGSM as we vary $\epsilon$, 2) compute the area under the curve (blue line), approximated by the area of the green region; 2) normalize the robustness value to the rectangular area formed by the *Ideal point* and *Nadir point*; 3) Ideal point is defined at 100% accuracy at pre-defined maximum $\epsilon$ value, and the nadir point is defined as the accuracy of random guessing at $\epsilon = 0$ (clean images).

Designing a measure/objective for robustness against adversarial robustness is an area of active research (e.g., (Carlini & Wagner, 2016)). For our purposes here, we present a possible measure here, illustrated in Figure 10. Using the fast gradient sign method (FSGM) presented by Goodfellow et al. (2014), this robustness objective progressively increases noise produced by FSGM. The $\epsilon$ axis in Figure 10 refers to the hyper-parameter in the FSGM equation,

$$\boldsymbol{x}' = \boldsymbol{x} + \epsilon \, \text{sign}(\nabla_{\boldsymbol{x}} J(\boldsymbol{x}, y_{true})),$$

where $x$ is the original image, $x'$ adversarial image, $y_{true}$ is true class label, and $J$ cross-entropy loss. Therefore, for this experiment, we seek to maximize two objectives, namely, classification accuracy and the robustness objective defined above.

The setup for the robustness experiment is as follows. For training we use 40,000 CIFAR-10 images from the official CIFAR-10 training data, 10,000 of which are reserved for validation. Each network is encoded with three phases (computational blocks). However, for this constrained experiment each phase has the same architecture i.e., we repeat the computational block. In each phase a maximum of size nodes may be active—where the computation at each node is as in the main paper: 3x3 convolution followed by ReLU and batch normalization. Each network is trained for 20 epochs with SGD on a cosine annealed learning rate schedule. The epsilon values used in the FSGM robustness calculation are [0.0, 0.01, 0.03, 0.05, 0.07, 0.1, 0.15]. As before, NSGA-Net initiates the search with 40 randomly created network architecture, and 40 new network architectures are created at each generation (iteration) via genetic operations (see main paper for details). The search is terminated at 30 generations.

### A.0.2 EXPERIMENT RESULTS

A clear trade-off between accuracy and robustness is shown in Figures 11 and 12c. NSGA-Net is capable of identifying a set of network architectures that provide an efficient trade-off between accuracy and robustness that would not have easily been obtained by hand. "Wide" networks (like ResNeXt (Xie et al., 2016) or Inception blocks (Szegedy et al., 2015)) appear to provide good accuracy on standard benchmark images, but are fragile to the FSGM attack. On the other hand, "deep" networks (akin to ResNet (He et al., 2016a) or VGG (Simonyan & Zisserman, 2015)) are more robust to FSGM attack, while having less accuracy. This phenomenon is illustrated with examples in Figures 12a and 12b, respectively. Furthermore, the skip connection provided by the modification we made to the original encoding proposed by Xie & Yuille (2017) appears to be critical in obtaining a network that is robust to adversarial attacks; see Fig. 13a.

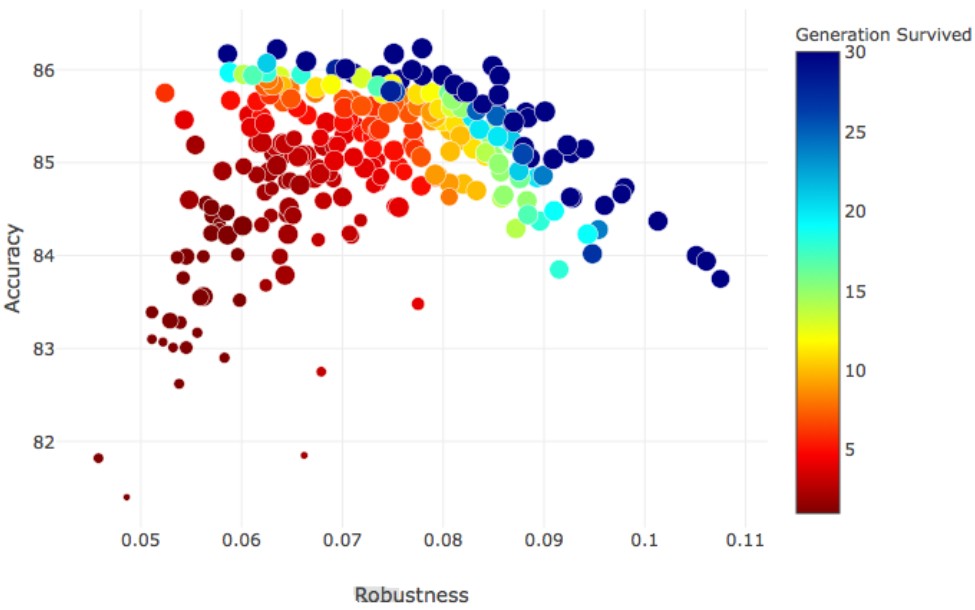

Figure 11: Trade-off frontier of the robustness experiments. Color indicates the generation (iteration) at which a network architecture is eliminated from the surviving parent population. The size of each point is proportional to the network architecture's number of trainable parameters. We note that networks for latter generations form the pareto front (dark blue points).

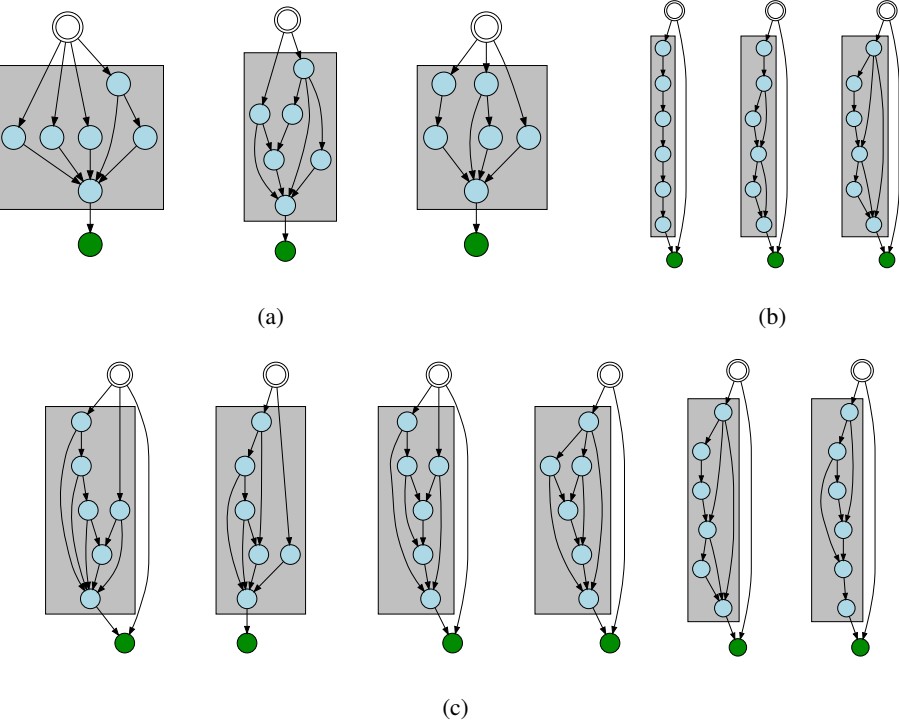

(a)

(b)

(c)

Figure 12: (a) Examples of the computational blocks discovered with high classification accuracy. For these networks, the mean accuracy and robustness objectives are 0.8543 and 0.0535, respectively; (b) Examples of the computational blocks discovered with high robustness against FGSM attack, the mean accuracy and robustness objectives are 0.8415 and 0.1036, respectively; (c) Examples of the computational blocks discovered along the pareto-front that provides an efficient trade-off between classification accuracy and adversarial robustness. They are arranged in the order of descending accuracy and ascending robustness.

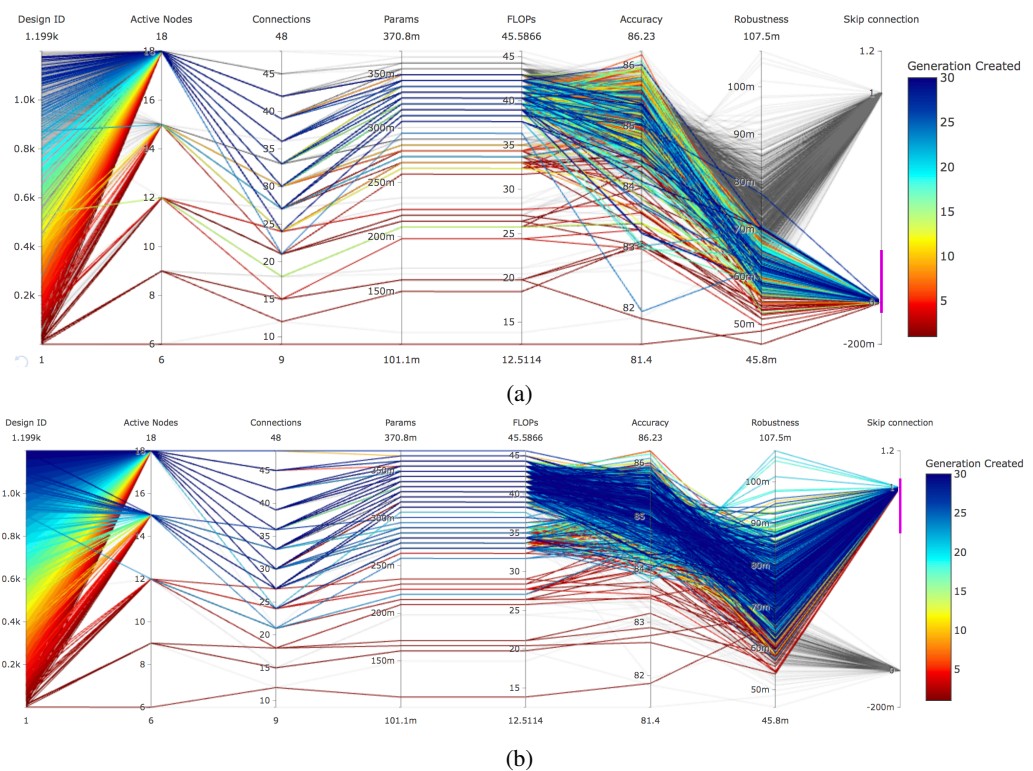

Figure 13: Parallel coordinate plots of the 1,200 network architectures sampled by NSGA-Net. Each line represents a network architecture, each vertical line is an attribute associated with the network. (a) Networks that have the skip connection bit inactive, we can see that none of them have good measurement on robustness against adversarial attacks. (b) Networks that have the skip connection bit active. This skip connection bit refers to the connection that goes past all computation within a phase, as a normal residual connection would. When the skip connection is active, the networks cover the full range of adversarial robustness.

## B    ADDITIONAL ABLATION EXPERIMENTS

To provide further analysis of the efficacy of the main components in NSGA-Net, we perform some additional ablation experiments. For these experiments, unlike the experiments in the main paper, we search for a single computational block that is then repeated as desired.

We compare NSGA-Net with four other alternative approaches; 1) random search i.e., 1) mutation only, without crossover, but with a higher mutation probability (maximum 5 bits change in each mutation operation, as opposed to 1 bit change in NSGA-Net; 2) No crossover but allow up to two mutations; 3) mutation only, without crossover, but with a small population size of four individuals, and 4) a one-point crossover followed by mutation with low mutation probability (maximum 1 bit change).

For all of the above variations we use the same experimental setup. We encode every network architecture with three phases and a maximum of six nodes in each of phase. We also restrict the networks to have the same architectures in all three phases, i.e., we repeat a block three times. Again, we split 40,000 images from the official CIFAR-10 training set for training and the remaining 10,000 images for validation. Each network architecture is then trained for 20 epochs with SGD method on a cosine annealing learning rate schedule. All ablation experiments share the same initial population to begin the search and all hyper-parameters associated to NSGA-Net, like population size, number of generations, etc., are the same unless specifically mentioned.

The results of our experiment are shown in Fig. 14a. We make the following observations: it is clear empirically that i) crossover is beneficial as the two strategies that use crossover out-perform the other three strategies; ii) mutation with high probability is harmful to the overall search. We then remove the restriction of having repeated phase architectures to study the effect of our proposed Bayesian learning based network architecture creation mechanism. The empirical results shown in Figure 14b indicate the improvements from non-repeated phase architecture as well as the Bayesian strategy.

## C    CROWDING DISTANCE VS. HYPERVOLUME-CONTRIBUTION

The goal of multi-objective optimization is to obtain a set of converged, diverse, and evenly distributed solutions along the pareto-front. Hypervolume (HV) is a popular metric to quantify these aspects (convergence and diversity) simultaneously. However, choosing points during a multi-objective search that maximize the HV metric may not necessarily result in a well-distributed trade-off frontier. Ishibuchi et al. (2018) have shown that algorithms driven purely by HV metric can lead to inferior outcomes (specifically, Figures 1 and 2 in (Ishibuchi et al., 2018) demonstrate the complexities of using a HV based approach). We have provided an example in Figure 15a to show the potential discrepancy between the HV metric value and the desired outcome i.e., obtaining a set of converged, diverse, and evenly distributed solutions along the pareto-front.

In addition, HV-contribution is not necessarily consistent with the HV metric depending on the number of objectives and concavity of the pareto-front. Finding a subset that maximizes the HV metric is an NP-hard process that involves solving a $N$ choose $k$ problem and in general a greedy solution to this search is a sub-optimal solution. A large body of the literature exists under the category of hypervolume subset selection problem, (Guerreiro et al., 2016; Ishibuchi et al., 2017). In short, using HV-contribution as opposed to CD will not ensure a better HV metric in the end nor a monotonic increase in the HV metric over the generations. We have provided an example in Figure 15b to show that HV-contribution can result in a worse HV metric value as compared to CD.

## D    RELATED WORK CONTINUED

As mentioned in the main paper, NAS has seen a methodological explosion in the past 2 years. Table 2 attempts to summarize the published and relevant methods in this area. A more complete listing, including unpublished methods, would be much longer. It is natural to group methods into EA and RL approaches, with some more exotic methods that fit into neither. The main motivation of EA methods is to treat structuring a network as a combinatorial optimization problem. EAs operate with a population that makes small changes (mutation) and mixes parts (crossover) of solutions to guide a search toward optimal solutions. RL, on the other hand, views constructing a network as a decision

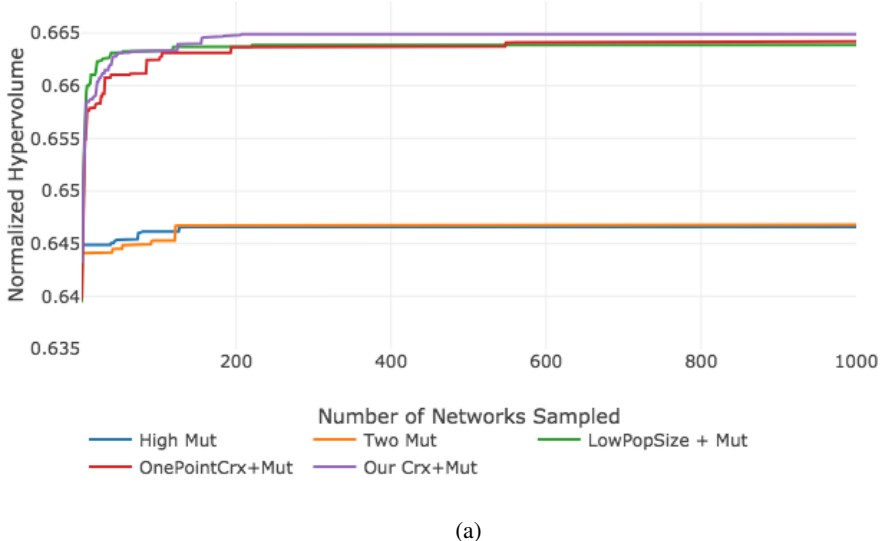

(a)

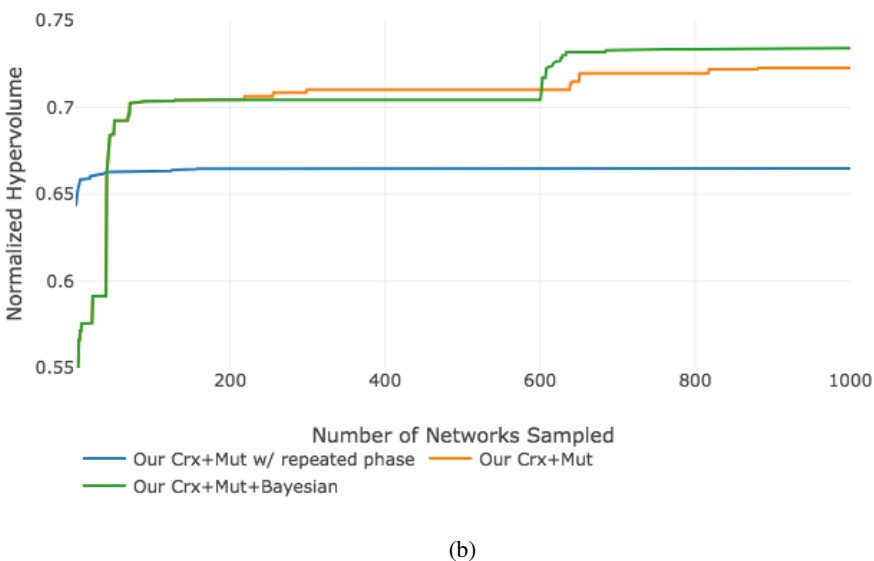

(b)

Figure 14: Median normalized hypervolume values over three runs of each strategy are compared. (a) We restrict networks to have the same phase architectures in all three phases. All strategies start with the same initial population, but different random seed for subsequent search operations. (b) We now allow each phase to have different architectures and we include the best strategy from (a) as a reference.

process. Usually, an agent is trained to optimally choose the pieces of a network in a particular order. We briefly review a few methods here.

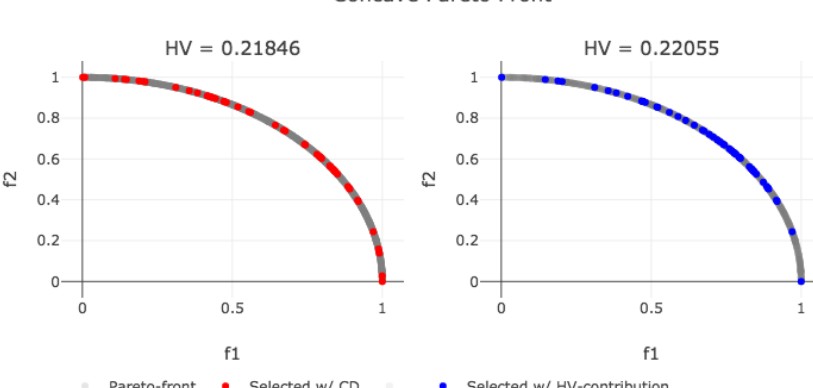

(a) A hypothetical two-objective-minimization problem with a concave pareto front. Both objectives along the axes are being minimized. 50 points were selected from 1,500 randomly sampled points from the pareto-front using crowding distance (CD) and HV-contribution. Observe that the solutions selected using CD more evenly span the pareto-front. It is clear that a better HV metric value need not necessarily come from a better distributed pareto-front.

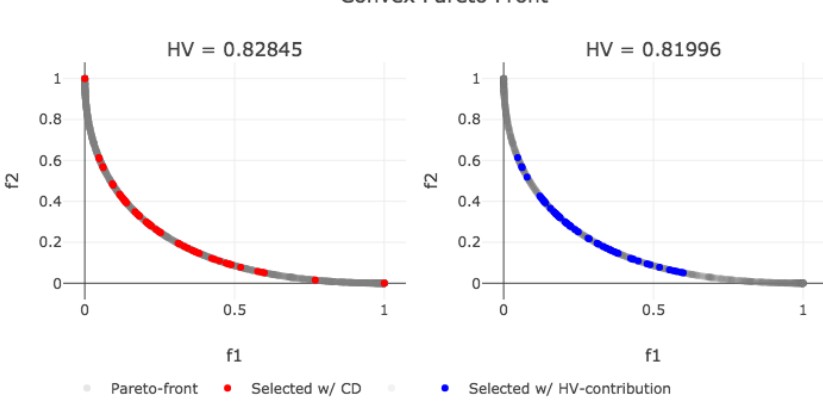

(b) A hypothetical two-objective-minimization problem with a convex pareto front. Both objectives along the axes are being minimized. 50 points were selected from 1,500 randomly sampled points from the pareto-front using crowding distance (CD) and HV-contribution. It's clear that HV-contribution need not necessarily ensure a better HV metric value as compared to selection with CD.

**Reinforcement Learning:** $Q$-learning (Watkins, 1989) is a widely popular value iteration method used for RL. The MetaQNN method (Baker et al., 2017) employs an $\epsilon$-greedy $Q$-learning strategy with experience replay to search connections between convolution, pooling, and fully connected layers, and the operations carried out inside the layers. Zhong et al. (Zhong et al., 2017) extended this idea with the BlockQNN method. BlockQNN searches the design of a computational block with the same $Q$-learning approach. The block that is then repeated to construct a network. This method allows for a much more general network and achieves better results than its predecessor on CIFAR-10.

A policy gradient method seeks to approximate some non-differentiable reward function to train a model that requires parameter gradients, like a neural network. Zoph & Le (2016) first applied this method in architecture search to train a recurrent neural network controller that constructs networks. The original method in (Zoph & Le, 2016) uses the controller to generate the entire network at once. This contrasts from its successor, NASNet (Zoph et al., 2017), which designs a convolutional and pooling block that is repeated to construct a network. NASNet outperforms its predecessor

and produces a network achieving state-of-the-art error rate on CIFAR-10. NSGA-Net differs from RL methods by using precise selection criteria—in fact, any EA method shares this characteristic. More specifically, networks are selected for their accuracy on a task, rather than an approximation of accuracy, along with computational complexity. Furthermore, the most successful RL methods search only a computational block that is repeated to create a network, NSGA-Net allows for search across computational blocks and combinations of blocks. Hsu et al. (2018a) extend the NASNet approach to a multiobjective domain to optimize a linear combination of accuracy and energy consumption. However, a linear combination of objectives has been characterized as suboptimal (Deb et al., 2000).

**Evolutionary Algorithms:** Designing neural networks through evolution, or *neuroevolution*, has been a topic of interest for some time, first showing popular success in 2002 with the advent of the neuroevolution of augmenting topologies (NEAT) algorithm (Stanley & Miikkulainen, 2002). In its original iteration, NEAT only performs well on comparatively small networks. Miikkulainen et al. (2017) attempt to extend NEAT to deep networks with CoDeepNEAT using a co-evolutionary approach that achieved limited results on the CIFAR-10 dataset. CoDeepNEAT does, however, produce state-of-the-art results in the Omniglot multi-task learning domain (Liang et al., 2018).

Real et al. (2017) introduced perhaps the first truly large scale application of a simple evolutionary algorithm. The extension of this method presented in (Real et al., 2018), called AmoebaNet, provided the first large scale comparison of EC and RL methods. Their simple EA searches over the same space as NASNet (Zoph et al., 2017) and has shown faster convergence to an accurate network when compared to RL and random search. Furthermore, AmoebaNet obtains the state-of-the-art results on CIFAR-10.

Conceptually, NSGA-Net is closest to the Genetic CNN (Xie & Yuille, 2017) algorithm. Genetic CNN uses a binary encoding that corresponds to connections in convolutional blocks. In NSGA-Net we augment the original encoding and genetic operations by (1) adding an extra bit for a residual connection, (2) introducing an encoding scheme for multi-resolution processing, and (3) introducing within-phase crossover. We also introduce a multi-objective based selection scheme. Moreover, we also diverge from Genetic CNN by incorporating a Bayesian network in our search to fully utilize past population history.

Evolutionary multiobjective approaches have been limited in past work. Kim et al. (2017) present an algorithm utilizing NSGA-II (Deb et al., 2000), however their method only searches over hyperparameters and a small fixed set of architectures. The evolutionary method shown in (Elsken et al., 2018) uses weight sharing through network morphisms (Wei et al., 2016b) and approximate morphisms as mutations and uses a biased sampling to select for novelty from the objective space rather than a principled selection scheme like NSGA-II (Deb et al., 2000). Network morphisms are also an important method in many approaches. They allow for a network to be "widened" or "deepened" in a manner that maintains functional equivalence. For architecture search, this allows for easy parameter sharing after a perturbation in a network's architecture.

**Other Methods:** Methods that do not subscribe to either an EA or RL paradigm have also shown success in architecture search. Liu et al. (2017) present a method that progressively expands networks from simple cells and only trains networks the best $K$ networks that are predicted to be promising by a RNN meta-model of the encoding space. Dong et al. (2018) extended this method to use a multiobjective approach, selected the $K$ networks based on their Pareto optimality when compared to other networks. Hsu et al. (2018b) also present a meta-model approach that generates models with state of the art accuracy. This approach may be ad hoc as no analysis is presented on how the progressive search affects the trade-off frontier. Thomas Elsken (2018) use a simple hill climbing method along with a network morphism (Wei et al., 2016b) approach to optimize network architectures quickly on limited resources. Chen et al. (2018b) combine the ideas of RL and EA. A population of networks is maintained and are selected for mutation with tournament selection (Goldberg & Deb, 1991). A recurrent network is used as a controller to learn an effective strategy to apply mutations to networks. Networks are then trained and the worst performing network in the population is replaced. This approach generates state of the art results for the ImageNet classification task. Chen et al. (2018a) show that an augmented random search approach to optimize networks for a semantic segmentation application. Finally, Kandasamy et al. (2018) present a Gaussian process based approach to optimize network architectures, viewing the process through a Bayesian optimization lens.

Table 2: Summary of relevant related work along with datasets each method has been applied to, objectives optimized, and the computational power used (if reported). Methods not explicitly named are presented as the author names. PTB refers to the Penn Treebank (Marcus et al., 1994) dataset. The Dataset(s) column describes what datasets the method performed a search with, meaning other datasets may have been presented in a study, but not used to perform architecture search. A dash represents some information not being provided. We attempt to limit the focus here to published methods, though some unpublished methods may be listed for historical contingency.

| | Method Name | Dataset(s) | Objective(s) | Compute Used |
|---|---|---|---|---|
| **RL** | Zoph and Lee (Zoph & Le, 2016) | CIFAR-10, PTB | Accuracy | 800 Nvidia K80 GPUs 22,400 GPU Hours |
| | NASNet (Zoph et al., 2017) | CIFAR-10 | Accuracy | 500 Nvidia P100 GPUs 2,000 GPU Hours |
| | BlockQNN (Zhong et al., 2017) | CIFAR-10 | Accuracy | 32 Nvidia 1080Ti GPUS 3 Days |
| | MetaQNN (Baker et al., 2017) | SVHN, MNIST CIFAR-10 | Accuracy | 10 Nvidia GPUs 8-10 Days |
| | MONAS (Hsu et al., 2018a) | CIFAR-10 | Accuracy & Power | Nvidia 1080Ti GPUs |
| | EAS (Cai et al., 2018) | SVHN, CIFAR-10 | Accuracy | 5 Nvidia 1080Ti GPUs 2 Days |
| | ENAS (Pham et al., 2018) | CIFAR-10, PTB | Accuracy | 1 Nvidia 1080Ti GPUs < 16 Hours |
| **EA** | CoDeepNEAT (Miikkulainen et al., 2017) | CIFAR-10, PTB | Accuracy | 1 Nvidia 980 GPU |
| | Real *et al.* (Real et al., 2017) | CIFAR-10, CIFAR-100 | Accuracy | - |
| | AmoebaNet (Real et al., 2018) | CIFAR-10 | Accuracy | 450 Nvidia K40 GPUs ~7 Days |
| | GeNet (Xie & Yuille, 2017) | CIFAR-10 | Accuracy | 10 GPUs 17 GPU Days |
| | NEMO (Kim et al., 2017) | MNIST, CIFAR-10 Drowsiness Dataset | Accuracy & Latency | 60 Nvidia Tesla M40 GPUs |
| | Liu *et al.* (Liu et al., 2018) | CIFAR-10 | Accuracy | 200 Nvidia P100 GPUs |
| | LEMONADE (Elsken et al., 2018) | CIFAR-10 | Accuracy | Titan X GPUs 56 GPU Days |
| | PNAS (Liu et al., 2017) | CIFAR-10 | Accuracy | - |
| | PPP-Net (Dong et al., 2018) | CIFAR-10 | Accuracy & Params/FLOPS/Time | Nvidia Titan X Pascal |
| **Other** | NASBOT (Kandasamy et al., 2018) | CIFAR-10 Various | Accuracy | 2-4 Nvidia 980 GPUs |
| | DPC (Chen et al., 2018a) | Cityscapes (Chen et al., 2014) | Accuracy | 370 GPUs 1 Week |
| | NAO (Hsu et al., 2018b) | CIFAR-10 | Accuracy | 200 Nvidia V100 GPUs 1 Day |

# E  ENCODING DETAILS

The overall architecture comprises of different phases (computational blocks), within each phase, the resolution of the information is maintained. Each phase comprises of a set of nodes (basic computational unit) which are operation or a sequence of operations to be performed on the inputs.

The maximum number of phases $n_p$ in an overall neural-network architecture is pre-specified. Resolution of a phase is determined by path encoding $\mathbf{x_p}$, while, the set of operations to be executed within a given phase is encoded in $\mathbf{x_o}$ vector. In the following sections, we discuss the *operation encoding* $\mathbf{x_o}$ and the *path encoding* $\mathbf{x_p}$ that are combined to create an entire network genotype $\mathbf{x} = (\mathbf{x_p}, \mathbf{x_o})$.

### E.1 OPERATION ENCODING: $\mathbf{x_o}$

We emphasize that this encoding is originally used in (Xie & Yuille, 2017). We present a small variation on that method with a different notation here. A *phase* is a computational block of the overall neural-network architecture. Thus, each phase is a convolution-neural-network by itself. We will first explain the operation-encoding for a phase, $\mathbf{x_o^{(i)}}$, where $i$ is the phase number/phase-id. The overall operation-encoding $\mathbf{x_o}$ is then generated by concatenating these phase-encodings $\mathbf{x_o^{(i)}}$, i.e. $\mathbf{x_o} = [\mathbf{x_o^{(1)}}, \mathbf{x_o^{(2)}}, \dots, \mathbf{x_o^{(n_P)}}]$, where $n_p$ is the number of phases.

The proposed encoding scheme is similar to the one presented in (Xie & Yuille, 2017), with a minor modification, wherein we append an extra bit to represent a direct connection from the *main input* to the *main output*. The resultant architecture is a directed graph where each *node* of the graph encapsulates following operations in sequence: *convolution (3×3), batch-normalization* and *ReLu*. Nodes are assigned with a *node-id*, which can be assume an integer value from 1 to $n$. Information in the neural network architecture is constrained to flow from a node with lower node-id to a node with higher node-id. Binary encoding is then generated as follows:

- Starting from node-2, each node has its corresponding substring of length $j - 1$ (where $j$ is the *node-id*, and $j \geq 2$).
- Elements of the binary substring of $j^{th}$ node can be represented by $b_i^{(j)}$, where $i = 1, \dots, j - 1$ and

$$b_i^{(j)} = \begin{cases} 1 & \text{if node } i \text{ is connected to node } j \\ 0 & \text{otherwise} \end{cases}$$

Thus, for a phase with $n$-nodes, $n \times (n - 1)/2$ bits are required to generate the encoding. It is possible for some nodes to have only outflow of the information and no input, while some nodes can have only inflow of the information and no output connection. These types of nodes are connected to the *main-input-node* and the *main-output-node*, respectively. Nodes which are devoid of any inputs and outputs (hanging nodes), are expelled from the final architecture. An extra bit is appended to represent a residue connection from the main-input-node to the main-output-node.

Finally, operation encoding $\mathbf{x_o}$ of the overall-network is generated by concatenating encodings $(x_o^{(j)})$ of each phase. If the maximum number of nodes a phase $j$ can have is $n_n^{(j)}$, and the maximum number of phases allowed in the final architecture is $n_p$, then the length of $\mathbf{x_o}$ will be $\sum_{i=1}^{n_p} n_n^{(i)} = L_s$.

The search-space of $\mathbf{x_o}$ thus comprises of $2^{L_s}$ binary-strings. This search-space however has redundancy as multiple sub-strings/genomes can decode to generate identical directed-graph. Since the training of a convolution-neural-network is a computationally expensive task, it is necessary to avoid training of the same CNN architecture which is represented by a different encoding. To achieve this, we have devised an approximate duplicate-check algorithm, described in Section E.3.

### E.2 PATH ENCODING $\mathbf{x_p}$

As mentioned before, the main neural-network is partitioned into different phases and each phase operates at a particular image-resolution. A $n_p$-dimensional vector, $\mathbf{x_p}$ (where $n_p$ is the maximum number of phases a neural-network architecture can have), is used to store the information about the resolution of each phase.

In NSGA-Net, the path encoding, $\mathbf{x_p} = (\mathbf{x_p^{(1)}}, \mathbf{x_p^{(2)}}, \dots, \mathbf{x_p^{(n_P)}})$, is a $n_p$-dimensional integer vector whose entries are in the range $[-r, r]$, where $r$ is the original input information resolution (e.g. for CIFAR-10, $r = 32$) and $n_p$ is the number of phases (computational blocks) in the network. Each $\mathbf{x_p}^{(i)}$ indicates the stride value for the pooling operation after $i^{\text{th}}$ phase, where positive encodes

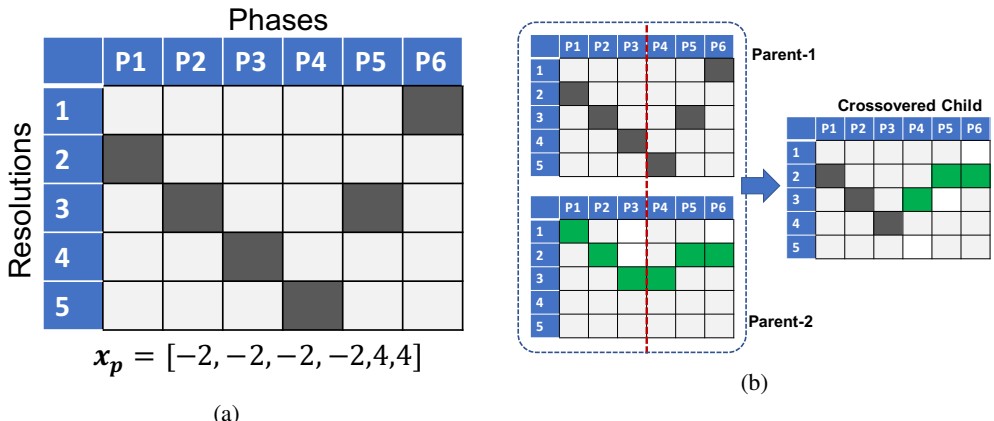

Figure 16: (a) An example of the path encoding in NSGA-Net. (b) An example of path crossover in NSGA-Net.

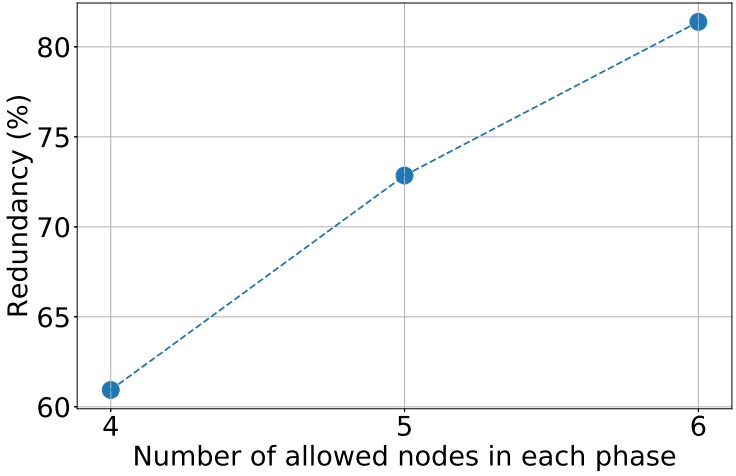

Figure 17: Increase in operation redundancy as node count increases.

up-sampling, negative encodes down-sampling and zero encodes no pooling operation. See 16a for a pictorial representation of this. See 16b for an example of path-based crossover.

### E.3 DUPLICATE CHECKING AND REMOVAL

Due to the directed acyclic nature of our encoding, redundancy exists in the search space defined by our coding, meaning that there exist multiple encoding stings that decode to the same network architecture. Empirically, we have witnessed the redundancy becomes more and more severe as the allowed number of nodes in each phase's computational block increase, as shown in Figure 17.

Since the training of a deep network is a computationally taxing task, it is essential to avoid the re-computation of the same architecture. In this section, we will provide with an overview of an algorithm we developed to quickly and approximately do a *duplicate-check* on genomes. The algorithm takes two genomes to be compared as an input, and outputs a *flag* to indicate if the supplied genomes decode to same architecture.

In general, comparing two graphs is NP-hard, however, given that we are working with Directed Acyclic Graphs with every node being the same in terms of operations, we were able to design an

efficient network architecture duplicate checking method to identify most of the duplicates if not all. The method is built on top of simply intuition that under such circumstances, the duplicate network architectures should be identified by swapping the node numbers. Examples are provided in Figure 18. Our duplicates checking method first derive the connectivity matrix from the bit-string, which will have positive 1 indicating there is an input to that particular node and negative 1 indicating an output from that particular node. Then a series row-and-column swapping operation takes place, which essentially try to shuffle the node number to check if two connectivity matrix can be exactly matched. Empirically, we have found this method performs very efficiently in identifying duplicates. An example of different operation encoding bit-strings decode to the same network phase is provided in Figure 18.

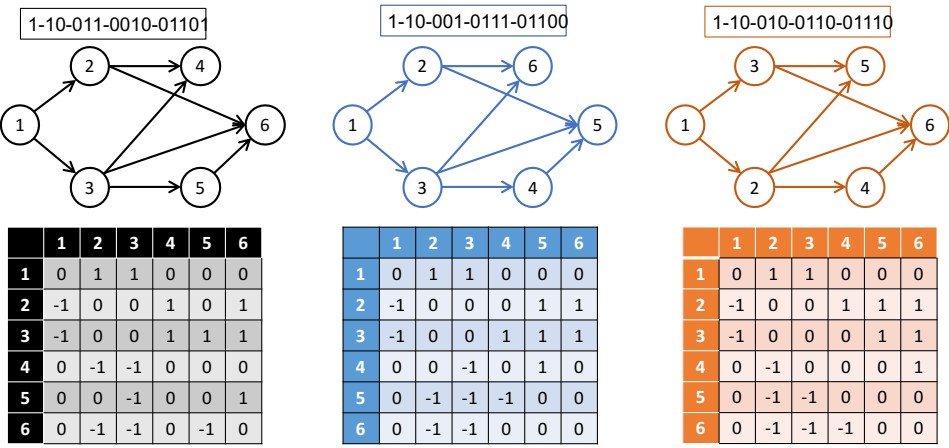

Figure 18: Examples of different encoding bit strings that decode to the same network computation block.

## F  NETWORK ARCHITECTURE COMPLEXITY ESTIMATION

We argue that the choice of inference time or number of parameters as proxies for computational complexity are sub-optimal and ineffective in practice. In fact, we initially considered both of these objectives. We concluded from extensive experimentation that inference time cannot be estimated reliably due differences and inconsistencies in computing environment, GPU manufacturer, and GPU temperature etc. Similarly, the number of parameters only relates one aspect of computational complexity. Instead, we chose to use Multiply-Adds (FLOPs) for our second objective. The following table compares the number of active nodes, the number of connections, the total number of parameters and the multiply-adds over a few sampled architecture building blocks. See Table 3 for examples of these calculations.

## G  EXPERIMENTAL RESULTS CONTINUED

Due to space constraints in the main paper, we present more interesting results here to show how NSGA-Net performs in comparison to other state-of-the-art methods.

### G.1  REGRESSION TASK DETAILS

In this section, we demonstrate another example of using NSGA-Net to find a set of efficient trade-off network architectures for object alignment task. We use the CMU-Car dataset described in (Boddeti et al., 2013). The CMU-Car dataset is an object alignment task containing around 10,000 car images in different orientations and environments.

Similarly to the classification example, we again use an 80/20 train/validation split from the original training set and the testing set contains only images of occluded cars and the training does not. For both tasks during the search *the testing data is not touched until the search concludes*. For

Table 3: Network examples comparing the number of active nodes, number of connections, number of parameters and number of multiply-adds.

| Networks | Nodes | Conn. | Params. | Multiply-adds |
|---|---|---|---|---|
| | 3 | 4 | 113 K | 101 M |
| | 4 | 6 | 159 K | 141 M |
| | 4 | 7 | 163 K | 145 M |
| | 5 | 9 | 208 K | 186 M |
| | 5 | 10 | 216 K | 193 M |
| | 6 | 13 | 265 K | 237 M |

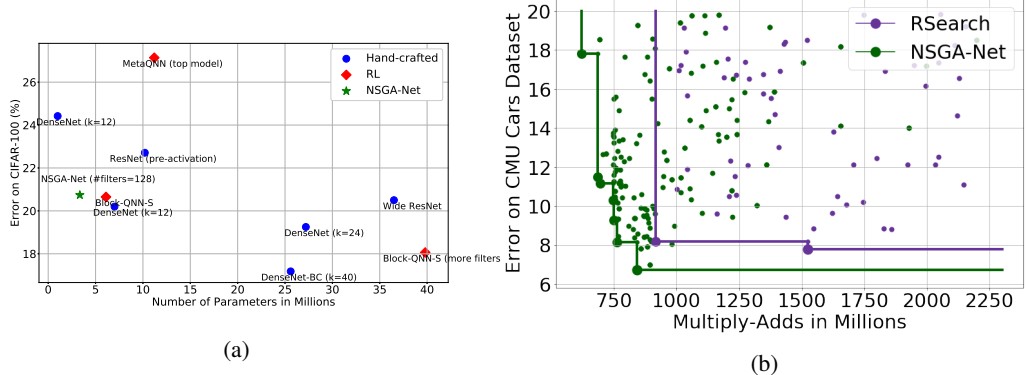

(a)

(b)

Figure 19: (a) CIFAR-100 results comparing hand-crafted neural networks architectures and neural architecture search found networks in the criterion space. Note that NSGA-Net is a non-dominated point. (b) Trade-off frontiers comparing NSGA-Net and RSearch on CMU-Cars dataset.

back-propagation in object alignment, we use RMSProp (Ruder, 2016) again with a cosine annealing learning rate (Loshchilov & Hutter, 2016). Our initial learning rate is 0.00025 and we train for 20 epochs which takes about 50 minutes on a single NVIDIA 1080Ti GPU.

## G.2 RESULTS

**Classification task**: Firstly, for the classification task on CIFAR-10 dataset, a more comprehensive comparison with state-of-art methods including both hand-crafted and search-generated is provided in Table 4. And examples of the efficient trade-off network architectures found by NSGA-Net on CIFAR-10 are provided in Figure 20. The reported NSGA-Net architecture in Table 4 with 128 filters is also trained on CIFAR-100, and the resulting performance compared to other methods is shown in Figure 19a.

**Regression task**: During the NSGA-Netsearch we fix the operation $\mathbf{x_o}$ to be the residual unit described in the original Hourglass paper (Newell et al., 2016). We then search over the path $\mathbf{x_p}$. We stack the hourglass twice during training and train for 20 epochs. We use a smaller population of 20 in the regression task. Once the search is done, we increase the number of filters and train

Table 4: NSGA-Net results (error rate) compare with state-of-the-art methods on CIFAR-10 (C-10) and CIFAR-100 (C-100) datasets.

| | Methods | Param. | Multiply -Adds | Error C-10 | C-100 | # Models Sampled | GPU Days | GPU Model |
|---|---|---|---|---|---|---|---|---|
| Hand Crafted | VGG (Simonyan & Zisserman, 2015) | - | - | 7.25 | - | - | - | - |
| | ResNet (He et al., 2016a) | 1.7M | 253M | 6.61 | - | - | - | - |
| | Wide ResNet (Zagoruyko & Komodakis, 2016) | 36.5M | 5953M | 4.17 | 20.50 | - | - | - |
| | ResNet (pre-activation) (He et al., 2016b) | 10.2M | - | 4.62 | 22.71 | - | - | - |
| | DenseNet (k = 12) (Huang et al., 2017) | 7.0M | - | 4.10 | 20.20 | - | - | - |
| | DenseNet (k = 24) (Huang et al., 2017) | 27.2M | - | 3.74 | 19.25 | - | - | - |
| | DenseNet-BC (k = 40) (Huang et al., 2017) | 25.6M | - | 3.47 | 17.18 | - | - | - |
| RL | MetaQNN (top model) (Baker et al., 2017) | 11.2M | - | 6.92 | 27.14 | 2,700 | 100 | |
| | Block-QNN-S (Zhong et al., 2017) | 6.1M | - | 4.38 | 20.65 | 10,800 | 96 | Titan X |
| | Block-QNN-S more filters (Zhong et al., 2017) | 39.8M | - | 3.54 | 18.06 | 10,800 | 96 | Titan X |
| | NASNet-A (6 @ 768) (Zoph et al., 2017) | 3.3M | - | 3.41 | - | 45,000 | 2,000 | P100 |
| | NASNet-A (7 @ 2304) (Zoph et al., 2017) | 27.6M | - | 2.97 | - | 45,000 | 2,000 | P100 |
| | MONAS (Hsu et al., 2018a) | - | - | 4.34 | - | - | - | 1080Ti |
| EA | GeNet v2 (Xie & Yuille, 2017) | - | - | 7.10 | - | - | - | Titan X |
| | CoDeepNEAT (Miikkulainen et al., 2017) | - | - | 7.30 | - | 7,200 | - | - |
| | Hierarchical(Liu et al., 2018) | - | - | 3.63 | - | 7,000 | 300 | P100 |
| | AmoebaNet-A (6, 36) (Real et al., 2018) | 3.2M | - | 3.34 | - | 20,000 | 3,150 | - |
| | AmoebaNet-B (6, 128) (Real et al., 2018) | 34.9M | - | 2.98 | - | 20,000 | 3,150 | - |
| | PPP-Net (Dong et al., 2018) | 11.39M | 1364M | 4.36 | - | - | - | Titan X |
| | RSearch w/ our encoding | 3.3M | 1247M | 3.86 | 21.97 | - | - | - |
| | NSGA-Net(#filters = 128) | 3.3M | 1290M | 3.85 | 20.74 | 1,200 | 8 | 1080Ti |
| | NSGA-Net(#filters = 256) | 11.6M | 4534M | 3.72 | 19.83 | 1,200 | 8 | 1080Ti |

for more epochs. Our best architecture obtains 8.64% error (Table 5). This is 1% worse than the state-of-the-art method, however at the cost of half the parameters, which may be desirable in some applications. The trade-off frontier achieved by NSGA-Net compared to uniform random sampling from our path encoding, $\mathbf{x_p}$, is provided in Figure 19b.

Table 5: Results of the regression search compared to the original Hourglass method (Newell et al., 2016).

| Method | Param. | Multiply-Adds | Error |
|---|---|---|---|
| Hourglass(Newell et al., 2016) | 3.38M | 3613M | 7.80 |
| NSGANet-A | 1.53M | 2584M | 8.66 |
| NSGANet-B | 1.61M | 2679M | 9.99 |
| NSGANet-C | 1.61M | 2663M | 8.64 |

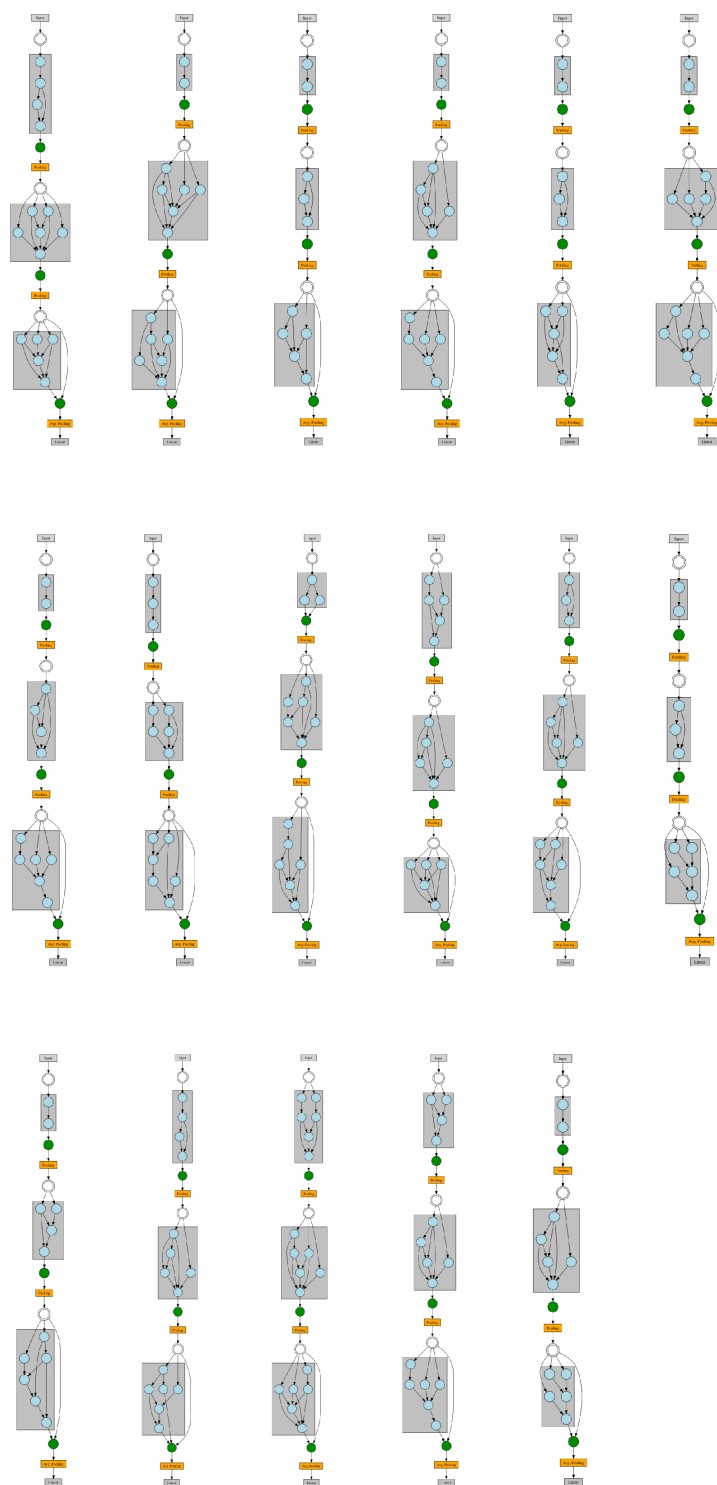

Figure 20: Set of networks architectures on the final trade-off front discovered by NSGA-Net.

