# OpenReview forum: "NSGA-Net: A Multi-Objective Genetic Algorithm for Neural Architecture Search"
_ICLR.cc/2019/Conference_

### Official Review · AnonReviewer2 · 2018-10-17
**Poorly justified approach**

**Rating:** 5
**Confidence:** 4

**Review:**

After rebuttal, I adapted the score. See below for original review.
--------------------------------------------


The authors implement a two-stage multi-objective optimization scheme to optimize neural network architectures with several conflicting goals.
I can not accept the paper in its current form.

In short, I have the following main criticisms:
1. use of crowding distance(CD) instead of hypervolume-contribution.
CD is not consistent with the HV estimator, especially CD might remove solutions that have a large HV-contribution and thus HV will not increase monotonically. The effect is even visible in Figure 8c) as in iteration 22, HV is decreasing as crowding distance removes a good offspring. In short: Crowding distance should not be used as long as the number of objectives does not prohibit computing the HV-contribution.

2. No good justification of BN. It is unclear to me why BN should be used instead of more iterations at stage 1. In 4.4 BN is only compared to the uniform initialization, but this comparison has no meaning given that we already have an optimized front that improved on the uniformly sampled distribution. To be honest, the samples shown from BN do not look very convincing as a lot of very poor architectures are created.

A proper comparison would be comparing the 2-step approach with only the first step and the same budget. Then we could compare samples from both distributions (either sampling from the front using mutation/crossover or sampling from BN). Also we would have a fair comparison of the obtained fronts and HV-values.

3. Ablation study cross-over
I am not convinced by the results presented. The paper says this is a "small scale" study but does not give the number of iterations/samples. It is clear that in the setup of the mutation operator cross-over might help, simply because it can change many more connections in a single iteration than mutation alone, which is limited to max 1 change. Allowing up to two mutations and no crossover could already proof to be better (orsmaller size of offspring population, see below)


Smaller concerns:

1. The results suggest that the uniform distribution might not be tuned well, as it only covers the "expensive" networks but not the "cheap" networks. A better initialization scheme that covers the x-axis better might already show vastly different results. As the Flop-objective is cheap to compute and does not require simulation, one could expect to tune this offline before initialization.

2. No handling of Noise.
During optimization, the chosen starting point and SGD algorithm will introduce noise into the process. Thus, the final test accuracy will be noisy. As an elitist dominance scheme is used, one might easily end up with an architecture that has a large variance when trained, i.e. when performing a final training pass on the full dataset, the performance might be very different. Moreover, the algorithm might stop convergence towards the true pareto front as it is held back by noisy "good" results. This should be discussed in the paper

3. A single-offspring approach might be better than sampling a full population (or offspring size in the order of parallel instances one can expend to run). 40 sounds excessive given that the sampling distribution is only improved through selection and given that the pareto front approximation appears to include less than 40 elements. This might also affect the results in the ablation study for cross-over: more iterations with reduced offspring size allows for more mutations of successful offspring.

4. Some unclear or wrong wordings:
page 4: "As a consequence[...] the best solution encountered [...] will always be present in the final population. " What do you consider "best" in a 2-objective problem? Do you mean: the best in each objective?
page 6, footnote1: this is not true. even without crossover the selection operator ties the solutions together, an offspring has to beat any point in the population, not necessarily its direct parent.

5. Figure 8a) does not include the state of the art result for CIFAR10, see for example
http://rodrigob.github.io/are_we_there_yet/build/classification_datasets_results.html#43494641522d3130

---

> ### Author Response · Authors · 2018-11-29
> **Responses to review comments**
>
>
> Main concerns:
>
> 1, The review’s criticism of the use of CD, as opposed to HV-contribution, is incorrect from the following aspects, more details and examples are provided in Appendix C of the revised submission.
>
> The review’s claim that using CD for selecting off-springs results in a non-monotonic increase in the HV metric over iterations. We argue the same is true with the HV-contribution for selecting off-springs, as suggested by the review. Finding a subset of solutions (greedily) from the non-dominated set is a combinatorial problem (N choose k), and in general, selection based on each solution’s HV-contribution is not the optimal solution to the problem. In Appendix C, we provide an example to show that a subset selected using CD can have a higher HV metric value than a subset selected using HV-contribution.
> Additionally, when the performance need to be assessed under multiple competing objectives, there is no single metric, including HV, that perfectly characterizes both convergence and diversity of the obtained solution set. Researchers have shown that algorithms solely driven by HV metric maximization can result in partial pareto-front being completely excluded [Ishibuchi et al. (2018)]. Appendix C provides an example to show that given a set of non-dominated points, a subset with higher HV metric value can be less evenly distributed as compared to another subset with lower HV metric value.
>
> Moreover, fundamentally, the goal of multi-objective optimization is NOT to improve the HV metric (or any other performance metric), instead, is to obtain a set of converged, diverse and evenly distributed solutions along the pareto-front.
>
> 2, BN is an effective method to identify the relationships between the phases. Examples are provided in these Bayesian Optimization Algorithm (BOA) papers [Pelikan et al. (1999), Pelikan et al. (2005)]. In the context of network architecture, we believe that the optimal phase (computational block) structure at later phases depends on the structure of the previous phases.
>
> The genetic operations in the exploration stage aim to find good computational block structures that preserve efficient trade-offs between objectives by inheriting common substructures between parents via crossover and injecting diversity via mutation. After evaluating a number of computational blocks (by the end of exploration), we now want to search for better ways to connect these blocks to form the entire network structure. The BN helps exploit the relations between promising combinations of the blocks.
>
> As suggested by the review, we ran an experiment that compares NSGA-Net with only genetic operations and NSGA-Net with both genetic operations and BN. In both setups, we use the same total search budget. The median hypervolumes from three runs of each setting are provided in Figure 14(b) in the Appendix B. The empirical results show that NSGA-Net is able to achieve a better hypervolume metric value with BN.
>
> 3, The “small scale” crossover experiment in the paper uses a population of size 20 and 5 iterations as the total search budget.
>
> We have revised our ablation study and more details are provided to address the concerns raised by the review. We implemented and evaluated all the methods suggested by the review: i) mutation alone with higher probability of mutation and ii) allowing up to two mutations with no crossover. The empirical results in Appendix B (Figure 14(a)) clearly shows that mutation alone is worse than crossover followed by mutation with low probability for our NAS setup.
>
> Conceptually, we argue that with crossover is better than mutation alone with large probability of mutation. Because 1) with large mutation probability, the search method will tend to behave similarly to random search as a large portion of the variables are randomly perturbed at every iteration; 2) mutation has no respect for variable linkages, in the context of neural architecture, crossover is capable of exchanging the sub-structures between two architectures and in the meantime maintaining the sub-structures locally unchanged.
>
> With an unlimited search budget, both with crossover and without crossover will likely to result in similar results. However, with limited search budget, our experiments suggest that using crossover yields a better performance in comparison to not using crossover. Figure 12(a) in Appendix B shows the results of our experiment.
>
>
> References:
> Ishibuchi,H., Imada, R., Setoguchi, Y., and Nojima, Y. How to specify a reference point in hypervolume calculation for fair performance comparison. Evol. Comput. 2018.
>
> Pelikan, M., Goldberg, D.E. and Cantu-Paz, E., 1999, July. BOA: The Bayesian optimization algorithm. In Proceedings of the 1st Annual Conference on Genetic and Evolutionary Computation.
>
> Pelikan, M., 2005. Hierarchical Bayesian optimization algorithm. In Hierarchical Bayesian Optimization Algorithm(pp. 105-129). Springer, Berlin, Heidelberg.

---

> ### Author Response · Authors · 2018-11-29
> **Responses to review comments**
>
>
> Small concerns:
> 1, The choice of FLOPS as the second objective for our experiments is perhaps does not showcase the full utility of multi-objective NAS, since the “Flop-objective is cheap to compute and does not require simulation, one could expect to tune this offline before initialization”
>
> For a different choice of the second objective “a better initialization”, as suggested by the review, may not be feasible. For example, consider robustness against adversarial attack as the second objective. “A better initialization” is not readily apparent in this case. An example of NSGA-Net applied to find neural architectures for classification accuracy (first objective) and robustness against adversarial attack (second objective) is provided in Section A in Appendix.
>
> 2, We agree with the review that noisy evaluations could potentially affect elitist dominance schemes. However, all NAS methods potentially suffer from the same problem.
>
> 3, We run the ablation study suggested by the review, where we reduced the population size from 40 to 4 and ran for 300 generations instead of 30. The empirical result, shown in Figure 14 (a) in Appendix B, suggests that running with low population size and longer generations reaches reasonable performance but still worse than evolving large population for fewer generations. The outcome of this approach is, however, critically dependent on the initialization. For other tasks, such as, robustness to adversarial attacks, such an initialization is not readily apparent.
>
> 4, We revised the language to reflect the review’s suggestions.

---

### Official Review · AnonReviewer1 · 2018-11-02
**The proposed method is interesting and promising approach, but the contribution is not clear.**

**Rating:** 5
**Confidence:** 3

**Review:**


- Summary
This paper proposes an evolutionary-based method for the multi-objective neural architecture search, where the proposed method aims at minimizing two objectives: an error metric and the number of FLOPS. The proposed method consists of an exploration step and an exploitation step. In the exploration step, architectures are sampled by using genetic operators such as the crossover and the mutation. In the exploitation step, architectures are generated by a Bayesian Network. The proposed method is evaluated on object classification and object alignment tasks.

- Pros
  - The performance of the proposed method is better than the existing multi-objective architecture search methods in the object classification task.
  - The effect of each proposed technique is appropriately evaluated.

- Cons
  - The contribution of the proposed method is not clear to me. The proposed method is compared with the existing multi-objective methods in terms of classification accuracy, but if we focus on that point, the performance (i.e., error rate and FLOPs) of the proposed method is almost the same as those of the random search judging from Table 4. It would be better to compare the proposed method to the existing multi-objective methods in terms of classification accuracy and other objectives.
  - This paper argues that the choice of the number of parameters is sub-optimal and ineffective in terms of computational complexity. Please provide more details about this point. For example, what is the drawbacks of the number of parameters, what is the advantages of FLOPs for multi-objective optimization?
  - Please elaborate on the procedure and settings of the Bayesian network used in this paper.
  - It would be better to provide discussions of recent neural architecture search methods solving the single-objective problem.

---

> ### Author Response · Authors · 2018-11-28
> **Responses to review comments**
>
> Below we address each of the concerns raised by the review.
>
> - The contribution of our paper is that we present an EA-based neural architecture search framework, named as NSGA-Net, that’s designed to find a competitive neural network architecture under single-criterion requirement or a set of neural network architectures that are representative to show the trade-offs between the multiple competing criteria. We argue that NSGA-Net is flexible and effective.
>
> The flexibility of NSGA-Net can be visualized from the following aspects:
> 1) NSGA-Net is capable of handling different tasks.
>     (a) An example of NSGA-Net applied to find neural architectures for CIFAR-10 images
>           classification is provided in Figure 8 and Table 1.
>     (b) An example of NSGA-Net applied to find neural architectures for objective alignment task
>          (CMU-Cars dataset) is provided in Section G.1 in Appendix.
>
> 2) NSGA-Net is capable of handling multiple objectives.
>     (a) An example of NSGA-Net applied to find neural architectures for classification accuracy and
>           network complexity is provided in Figure 8 and Table 1.
>     (b) An example of NSGA-Net applied to find neural architectures for classification accuracy and
>           robustness against adversarial attack is provided in Section A in Appendix.
>
> 3) NSGA-Net is capable of handling different datasets.
>      (a) Examples of NSGA-Net applied to MNIST, SVHN and CIFAR-10 are provided in Figure 9 (b) as
>            an ablation study. Due to space limit constraint, we are only able to show the hypervolume
>            comparison in the paper.
>
> The effectiveness of NSGA-Net can be visualized from the following aspects:
> 1) NSGA-Net finds network architectures with similar performance (in terms of both classification accuracy and network complexity) as compared to both hand-crafted network architecture (like ResNet, DenseNet) and search-assisted network architectures (like NASNet). See Figure 8 for details.
> 2) The overall search expense of NSGA-Net is significantly lower than other RL-based or EA-based neural architecture search methods. See Table 2 and Table 4 in Appendix for details.
>
>
> - FLOPs is a more comprehensive estimator for a network’s computational complexity, it is a composite measure of the number of parameters, the number of connections among the nodes and layers, and the operations taken place inside each node. For example, a DenseNet has a higher computational complexity as compared to a similar depth (number of parameters and layers) ResNet, which can be captured by FLOPs but not the number of parameters as those dense connections accumulates hardly any extra parameters, but require significantly more FLOPs.
>
>
> - The detailed procedure of our Bayesian network is as following:
> Prerequisites: i) our network architectures consist of three phases (computational blocks) and pooling operations in-between phases. And we are not repeating the phases. ii) we have collected an archive of evaluated network architectures from previous “exploration” step.
> Step 1: From the archive of all previously evaluated network architectures, we select the top 50% of the networks by non-dominated sorting followed by crowding-distance on both classification error and computational complexity.
> Step 2: we sample the “phase1” block from the distribution of frequency of occurrence of the “phase1” blocks. This occurrence frequency is estimated from the top 50% networks in the archive.
> Step 3: we sample the “phase2” block from the distribution of frequency of occurrence of the “phase2” blocks conditioned on the sampled “phase1” block in the previous step. This occurrence frequency is also estimated from the same top 50% networks in the archive.
> Step 4: similarly, we sample the “phase3” block from the distribution of frequency of occurrence of the “phase3” blocks conditioned on the sampled “phase2” block in the previous step. This occurrence frequency is also estimated from the same top 50% networks in the archive.
> Step 5: we then form our new network architecture by connecting “phase1”, “phase2”, and “phase3” with pooling operations.
>
>
> - This was already included in Appendix D of the submission.

---

### Official Review · AnonReviewer3 · 2018-11-03
**A combination of architecture search ideas**

**Rating:** 6
**Confidence:** 4

**Review:**

This paper proposes a search method for neural network architectures such that two (potentially) conflicting objectives: maximization of final performance and minimization of computational complexity can be pursued simultaneously. The motivation for the approach is that a principled multiobjective search procedure (NSGA-II) makes it unnecessary to manually find the right trade-off between two objectives, and simultaneously finds several solutions spanning the tradeoff. It is also capable of finding solutions from the concave regions of the Pareto-front. Multiobjective search for architectures has been explored in recent work, so the primary contribution of this paper is to show its utility in a more general and perhaps more powerful setting.

The paper is clearly written and is easy to understand, except that the parenthetical citations used appear to differ from the ICLR style and cause confusion. The authors delve into details of the approach though many aspects are from past work. I think that this makes the paper more self-contained and easy to understand, even if it makes the paper longer than the suggested length of 8 pages. I also found the comparisons and ablations shown in Figures 8 and 9 to be useful and informative.

However, based on the presented results on the CIFAR-10 dataset (which can be compared to past work), I am not convinced of the utility of multiobjective optimisation for architecture search. There are a few reasons for this:

1. The best architectures found by previous methods in the literature are already at a similar or better accuracy. It appears that NSGA-Net did not succeed in finding architectures that a) outperform past results with higher FLOPs, or b) match past results with fewer FLOPs. I understand that in principle, a benefit of NSGA-Net is that other solutions with lower accuracy and fewer FLOPs are also found simultaneously, but these models are not discussed or analysed much in detail. What precisely is the utility of the proposed method then? This consideration is also complicated by the next point.

2. For the evaluation in the paper, the network with the lowest accuracy is extrapolated — the number of filters in each layer are increased and the network is retrained. Is this procedure justified in general? How to know the best increasing factor?
Since lowering the computational cost is an objective of the search, changing the cost of an obtained solution does not seem principled.  Moreover, changing network sizes will affect any ordering of networks by accuracy since optimal hyperparameters for both optimization and regularization may change. In general, it is rather difficult to decouple hyperparameter search from architecture search.

3. A baseline that is missing in the paper is hyperparameter search, which can often yield very good performance for a given architecture. Tuning regularization in particular is often crucial. Since NSGA-Net trains 1200 networks, a comparable search would consider a known architecture e.g. Densenet and allocate 200 trials each to 6 architectures of different FLOPS (or 100 each to 12 architectures). How effective is this simple procedure at obtaining a good tradeoff front?

Due to these concerns, I am presently unconvinced by the results in this paper, though I think that in general multiobjective optimization of architectures should be a fruitful direction.

Minor question: Figure 9(b) indicates that experiments were also conducted on the SVHN and MNIST datasets. Why are these results not reported?

---

> ### Author Response · Authors · 2018-11-29
> **Responses to review comments**
>
>
> 1, The primary objective of our paper is to demonstrate the utility NSGA based multi-objective NAS, as opposed to outperforming previous methods. Our choice of FLOPS as the second objective does not fully showcase the utility of NSGA-Net. In Appendix A we consider another objective, robustness to adversarial attacks, as our second objective. To the best of our knowledge, hand tuning or automatically tuning networks that are robust to adversarial attacks has not been studied. Our experiments in Appendix A allow us draw some interesting conclusions, namely, 1) there exist trade-offs between classification accuracy and robustness to adversarial attack and NSGA-Net is capable of identifying a set of network architectures that provide an efficient trade-off between accuracy and robustness that would not have easily been obtained by hand; 2) “wide” networks (like ResNeXt (Xie et al., 2016) or Inception blocks (Szegedy et al., 2015)) appear to provide good accuracy on standard benchmark images, but are fragile to the FSGM attack. On the other hand, “deep” networks (akin to ResNet (He et al., 2016a) or VGG (Simonyan & Zisserman, 2015)) are more robust to FSGM attack, while having less accuracy; 3) the skip connection provided by the modification we made to the original encoding proposed by Xie & Yuille (2017) appears to be critical in obtaining a network that is robust to adversarial attacks.
>
> 2, The extrapolation step in NSGA-Net is adopted from other NAS approaches [Zoph et al., (2017); Real et al., (2018)], and it’s not a contribution of NSGA-Net. Having said that, the only reason of we use lower number of filters during the architecture search is for computational tractability. Moreover, in principle, the training process during architecture search cannot be exactly the same as when we train the best network, since only a subset of the training dataset is used during architecture search (to prevent over-fitting to the test set) while the entire training set is used for training the best network.
>
> We agree with the review that hyper-parameter tuning cannot be fully decoupled from architecture search. However, even without hyper-parameter tuning, NSGA-Net finds network architectures with competitive performance (in terms of both accuracy and complexity) to those hand-designed and hyper-parameter tuned networks (ResNet and DenseNet).
>
> 3, Tuning hyperparameters (such as learning rates, regularization parameters, etc.) cannot always tune complex objectives, such as robustness to adversarial attacks as shown in Appendix A. Furthermore, hyperparameter search is its own separate field of research and is complementary to the methods proposed here. Ideas from both fields could be combined, though hyperparameter tuning is not the goal of this study.
>
> 4, SVHN and MNIST datasets were only used for the ablation study on the effect of crossover. We did not run full experiments on those datasets.
>
>
> References:
> Zoph, B., Vasudevan, V., Shlens, J. and Le, Q.V., 2017. Learning transferable architectures for scalable image recognition. arXiv preprint arXiv:1707.07012, 2(6).
>
> Real, E., Aggarwal, A., Huang, Y. and Le, Q.V., 2018. Regularized evolution for image classifier architecture search. arXiv preprint arXiv:1802.01548.

---

> > ### Comment · AnonReviewer3 · 2018-12-07
> > **Post-rebuttal comments**
> >
> > 1. Unfortunately, despite thinking a lot about it, I am unconvinced that the paper currently demonstrates a clear utility of multi-objective architecture search. I think it is clear in theory that multi-objective search for architectures can have a utility. The question is: do we know how to do it right? The answer to this question is not clear from the experiments with FGSM robustness as the secondary objective either, although there are certainly good ideas here that my rating already takes into account.
> >
> > The FGSM attack is not considered a ‘serious’ attack at all under any threat model. This is known in recent work on adversarial attacks, but I can also point you to a direct comment from the author of FGSM [1].
> > In fact, I’d argue that from the current state of adversarial ML research, it is not clear at all whether architecture search would lead to models that are secure under realistic threat models. I’d be happy to be proven wrong by new results using NSGANet.
> >
> > It is likely that there are other objectives where the utility of multi-objective architecture search can be more clearly demonstrated, but at present these experiments are unconvincing to me.
> >
> > 2. When considering the complexity-performance trade-off, it does not appear correct to change the complexity of the network after optimisation (since the computational complexity and performance do not have a simple relationship). I can only comment on the paper under review, but perhaps this is not a major issue when the search method does not actively seek this trade-off. I’m sympathetic that computational requirements can be a hurdle in having the ‘right’ setup for the method.
> >
> > [1] https://openreview.net/forum?id=SkgVRiC9Km&noteId=rkxYnt8JpQ

---

### Meta-Review · Area_Chair1 · 2018-12-14

**Confidence:** 3
**Recommendation:** Reject

**Metareview:**

Pros:
- an explicitly multi-objective approach to neural architecture search
- multiple datasets
- ablation experiments

Cons:
- lack of baselines like hyperparameter search
- ill-justified increase in capacity after search
- ineffective use of the multiple objectives in assessment
- not clearly beating random search baseline

The reviewers adjusted their scores upward after the rebuttal, but serious concerns remain, and the consensus is still to (borderline) reject the paper.